

# Resolving the timescales of magmatic and hydrothermal processes associated with porphyry deposit formation using zircon U-Pb petrochronology

Simon J.E. Large[1][*][†], Jörn F. Wotzlaw[1], Marcel Guillong[1], Albrecht von Quadt[1], Christoph A. Heinrich[1,2]

[1]Department of Earth Sciences, Eidgenössische Technische Hochschule (ETH) Zurich, 8092 Zürich, Switzerland.
[2]Faculty of Mathematics and Natural Sciences, University of Zurich, 8006 Zürich, Switzerland.
[†]Current address: Department of Earth Sciences, Natural History Museum, Cromwell Road, London, SW7 5BD, UK

*Correspondence to*: Simon J.E. Large (s.large@nhm.ac.uk)

**Abstract.**

Understanding the formation of economically important porphyry-Cu-Au deposits requires the knowledge of the magmatic-to-hydrothermal processes that act within the much larger underlying magmatic system and the timescales on which they occur. We apply high-precision zircon geochronology (CA-ID-TIMS) and spatially resolved zircon geochemistry (LA-ICP-MS) to constrain the magmatic evolution of the magma reservoir at the Pliocene Batu Hijau porphyry-Cu-Au deposit. We then use this extensive dataset to assess the accuracy and precision of different U-Pb dating methods of the same zircon crystals.

Emplacement of the oldest pre- to syn-ore tonalite (3.736 ± 0.023 Ma) and the youngest tonalite porphyry cutting economic Cu-Au mineralisation (3.646 ± 0.022 Ma) is determined by the youngest zircon grain from each sample, which constrains the duration of metal precipitation to less than 90 ± 32 kyr. Overlapping spectra of single zircon crystallisation ages and their trace element distributions from the pre-, syn and post-ore tonalite porphyries reveal protracted zircon crystallisation together with apatite and plagioclase within the same magma reservoir over >300 kyr. The presented petrochronological data constrains a protracted early >200 kyr interval of melt differentiation and cooling within a large heterogeneous magma reservoir leading up to ore formation, followed by magma storage in a highly crystalline state and chemical and thermal stability over several 10s of kyr. Irregular trace element systematics suggest magma recharge or underplating during this final short time interval.

The comparison of high precision CA-ID-TIMS results with in-situ U-Pb geochronology data from the same zircon grains allows a comparison of the applicability of each technique as a tool to constrain dates and rates on different geological timescales. All techniques provide accurate dates with variable precision. Highly precise dates derived by the calculation of the weighted mean and standard error of the mean of zircon dates obtained by in-situ techniques can lead to significantly older suggested emplacement ages than those determined by high-precision CA-ID-TIMS geochronology. This lack in accuracy of the weighted means is due to the protracted nature of zircon crystallisation in upper crustal magma reservoirs, suggesting that standard errors should not be used as a mean to describe the uncertainty in those circumstances. Thus,



geologically rapid events or processes or the tempo of magma evolution are too fast to be reliably resolved by in-situ U-Pb

geochronology and require ID-TIMS geochronology.

**1 Introduction**

Zircon geochronology is widely applied to date geological events and constrain timescales of geological processes. Combined with zircon geochemistry it has improved our understanding of crustal magmatic systems, such as those forming

economically important magmatic-hydrothermal porphyry Cu-Au deposits. Advances in analytical techniques resulted in a shift from establishing the ages of magma emplacement or crystallisation to resolving the durations of magmatic and associated hydrothermal processes, such as magma accumulation or recharge, fractional crystallisation or hydrothermal ore formation and it has resulted in unprecedented information about the mechanisms and scales of magma ascent and storage in the Earth's crust (e.g. Vazquez and Reid, 2004;Chamberlain et al., 2014;Barboni et al., 2016;Bucholz et al., 2017).

Porphyry copper deposits provide successively quenched samples of magma extracted from large crustal-scale hydrous magma systems. They are therefore a critical source of information about the processes and rates of magma ascent, magma storage and fluid generation, bridging those of volcanism and pluton formation. The identification of the processes that lead to porphyry deposit formation (e.g. Rohrlach et al., 2005;Audétat et al., 2008;Richards, 2013;Wilkinson, 2013) and the timescales on which they can provide us with valuable information about arc magmatic processes but could also

potentially help in discriminating possibly fertile magmatic systems from ubiquitous infertile systems resulting in barren intrusions or volcanic eruptions.

Porphyry Cu-Au deposits commonly display clear field relationships of successive generations of porphyritic stocks or dikes, which were injected into subvolcanic and other upper-crustal rock sequences (Sillitoe, 2010). The injected porphyry magmas thus provide snapshots of the underlying, vertically and laterally extensive, magma reservoirs (e.g. Dilles,

1987;Steinberger et al., 2013). Cross-cutting relationships between veins and intrusive rocks suggest temporal overlap of hydrothermal alteration, ore mineralisation and porphyry emplacement (Proffett, 2003;Seedorff and Einaudi, 2004;Redmond and Einaudi, 2010). Strong hydrothermal alteration of the intrusive rocks associated with ore formation severely disturbs the geochemical information of most minerals and whole-rock compositions. While providing important insights into the hydrothermal history of a deposit (e.g. Roedder, 1971;Dilles and Einaudi, 1992;Landtwing et al., 2005;Cathles and Shannon,

2007;Seedorff et al., 2008;Large et al., 2016) it limits the investigation of the magma evolution, especially for the porphyries that are most intimately associated with ore formation. Zircon is a mineral that is unaffected by nearly all hydrothermal alteration and can thus provide unique information about the evolution of a magmatic system.

Recent advances in high-precision zircon geochronology by chemical abrasion - isotope dilution - thermal ionization mass spectrometry (CA-ID-TIMS: e.g. Mattinson, 2005;Bowring et al., 2011;McLean et al., 2011a;Condon et al.,

2015;McLean et al., 2015) now allow dating the porphyritic intrusions associated with ore formation with unprecedented



precision. The dramatically improved precision permits to constrain rapid events, such as individual porphyry emplacement and hydrothermal mineralization phases (<100 kyr: von Quadt et al., 2011;Buret et al., 2016;Tapster et al., 2016) that typically occur at the end of a longer-term period of volcanism and intrusive magma emplacement extending over several million years (e.g. Deino and Keith, 1997;Halter et al., 2004;Maksaev et al., 2004;Rohrlach et al., 2005;Lee et al., 2017).

The integration of the temporal and chemical information gained from zircon is referred to as zircon petrochronology and can yield time-calibrated information about magma chemistry, thermal evolution and crystallinity during zircon crystallisation in magmatic systems (e.g Schoene et al., 2012;Chelle-Michou et al., 2014;Samperton et al., 2015;Buret et al., 2016;Szymanowski et al., 2017).

Geological events and processes that require highest possible precision to be resolved, essentially rely on the

accuracy of the chosen analytical technique. Timescales for magmatic and hydrothermal processes involved in porphyry ore formation have been suggested based on in-situ U-Pb data (e.g. Garwin, 2000;Banik et al., 2017;Lee et al., 2017) and increasingly precise CA-ID-TIMS geochronology (e.g. von Quadt et al., 2011;Chelle-Michou et al., 2014;Buret et al., 2016;Tapster et al., 2016;Gilmer et al., 2017;Large et al., 2018). However, several studies applying multiple techniques on the same sample sets have resulted in differing dates (von Quadt et al., 2011;Chiaradia et al., 2013;Chelle-Michou et al.,

2014;Chiaradia et al., 2014;Correa et al., 2016). The discrepancy demands for a more detailed understanding of the precision and accuracy of the techniques and the statistical data treatment that are applied to derive a geological age. This is not only fundamental for resolving dates and rates of geological processes in porphyry research but equally affects magmatic dates and rates obtained by U-Pb geochronology.

For the present paper, we obtained a large dataset of zircon geochemistry and geochronology by laser ablation-

inductively coupled plasma-mass spectrometry (LA-ICP-MS) coupled to high-precision geochronology of the same zircon fragments/segments/crystals utilising chemical abrasion-isotope dilution-thermal ionization mass spectrometry (CA-ID-TIMS). The data from the world-class Batu Hijau porphyry Cu-Au deposit allows to resolve the chemical evolution and the changing physical state of the magma reservoir over time as well as the timescales of hydrothermal processes. In addition, previously published data on the same lithologies permit a critical comparison of two in-situ microanalytical methods

(SHRIMP data by Garwin (2000), LA-ICPMS presented here) with high-precision U-Pb CA-ID-TIMS geochronology (this study). This allows us to critically compare the effects of variable degrees of precision and of the statistical treatment of data on the resulting interpreted ages and it provides a mean to test the accuracy of the different techniques.

**2 Geological Background**

The Pliocene, island-arc hosted world-class porphyry deposit of Batu Hijau is located on Sumbawa island, Indonesia (Fig. 1), and it is one of the largest Cu and Au resources in the Southwest Pacific region (7.23 Mt Cu and 572 t Au: Cooke et al.,



2005). It is currently the only mined porphyry deposit in the Banda-Sunda volcanic arc, where Cu-Au porphyries are restricted to a narrow segment of the eastern Sunda-Banda arc from 115°E and 120°E (Fig.1). where Australian plate is being subducted since the Eocene (Hall, 2002).

The exposed islands of the Sunda-Banda arc are characterised by Late Oligocene to Early Miocene calc-alkaline basaltic to andesitic arc rocks that are overlain or intruded by a Late Miocene to Pleistocene calc-alkaline volcanic and plutonic rock suite ranging from basaltic to rhyolitic compositions (Hamilton, 1979;Hutchison, 1989). The magmatic arc hosts a variety of ore deposit types, including porphyry-Cu-Au deposits, high-, intermediate- and low-sulphidation epithermal deposits and a VMS-type deposit on Wetar (Fig. 1).

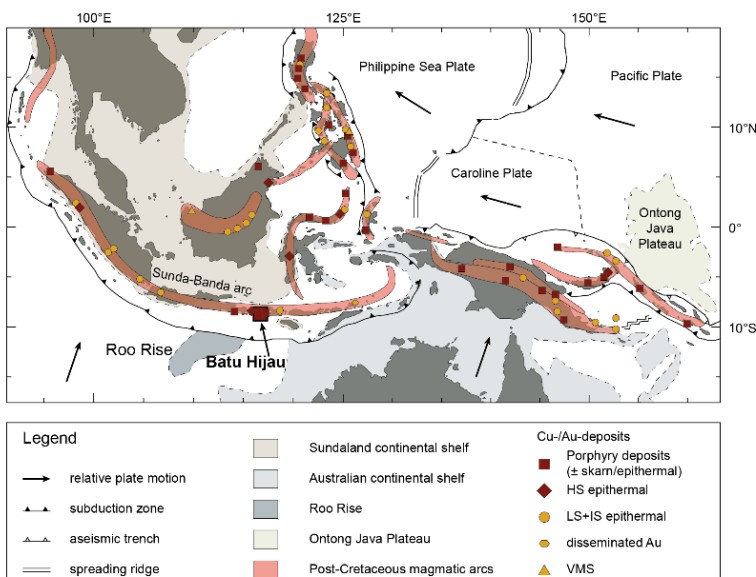

**Figure 1: Tectonic map of southeast Asia and the southwest Pacific. The Batu Hijau porphyry Cu-Au deposit (enlarged red square) is located on Sumbawa on the subduction related, magmatic Sunda-Banda arc within a small corridor between 110°E and 120°E that hosts several porphyry deposits. Arrows display plate motion relative to the Eurasian plate (Sundaland shield) but do not indicate velocity. Most deposit locations are from Garwin (2005).**

The geology of Sumbawa Island, hosting the Batu Hijau deposit, is dominated by Early Miocene to Holocene volcanic arc successions deposited on oceanic crust that is 14 – 23 km thick (Hamilton, 1979;Barberi et al., 1987). Thickened continental crust observed in most other porphyry-mineralized magmatic arcs and commonly considered a prerequisite for porphyry-Cu formation (Rohrlach et al., 2005;Lee and Tang, 2020) is lacking beneath Sumbawa (Garwin et al., 2005). The distribution of volcano-sedimentary units, intrusions and the current coastline of Sumbawa are controlled by a major arc-transverse, left-lateral oblique-slip fault zone (Arif and Baker, 2004;Garwin et al., 2005). The fault zone strikes





SSE-NNE about 30 km east of the Batu Hijau deposit coinciding with the north easterly projection of the Roo Rise oceanic
plateau (Fig. 1).

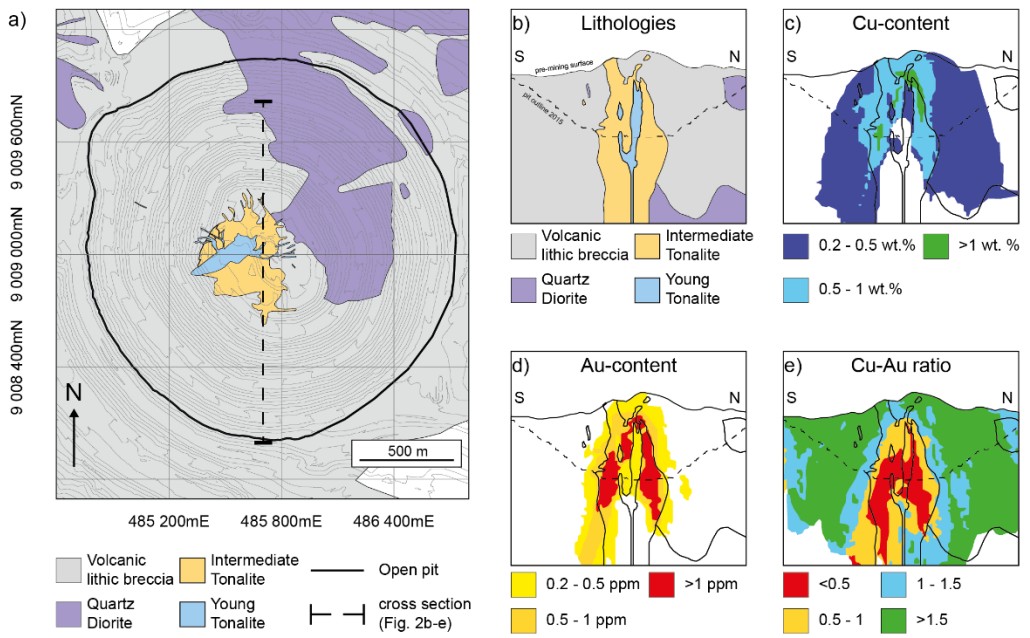

**Figure 2: Geological map (a) and north to south cross-sections with lithological information and grade contours (b-e)**
**of the open pit at Batu Hijau. The Intermediate and Young tonalite intruded into a Volcanic lithic breccia and the**
**equigranular quartz diorite (a, b). Note that the extent of the Old tonalite is not displayed but is included in the**
**Intermediate tonalite. Dashed line in a) is the N-S section displayed in b)-e). Thin grey lines in a) indicate mine-**
**benches. Cu- and Au-grades (c + d) are enveloped around the tonalites and a deep, central barren core. High-grade**
**Cu- and Au-mineralisation is cut by the Young tonalite. The ratios of Cu to Au (e) illustrate strong Au-enrichment**
**proximal to the Intermediate tonalite and Cu-dominated distal mineralization. Map, section and grades are based on**
**company information from May 2016.**

The hypabyssal stocks in the Batu Hijau district are intruded into an Early to Middle Miocene volcano-sedimentary
rock sequence (< 21 Ma based on biostratigraphy: Adams, 1984;Berggren et al., 1995) that reaches thicknesses of up to 1500
m in southwestern Sumbawa. The low $K_2O$, calc-alkaline, sub-volcanic intrusive rocks in the Batu Hijau district have
andesitic to quartz-dioritic and tonalitic compositions (Foden and Varne, 1980;Garwin, 2000) and were emplaced in several
pulses during the Late Miocene and Pliocene (Garwin, 2000). Over this multi-million year magmatic history, a continuous
geochemical evolution towards more fractionated lithologies is indicated by whole-rock chemistry and Fe-isotopic evidence



of the magmatic rock suite in the Batu Hijau district (Garwin, 2000;Wawryk and Foden, 2017). Within the Batu Hijau deposit, andesite porphyries and different quartz-diorite bodies are the earliest recognized stocks, whereas three tonalite porphyries are the youngest exposed intrusions (Clode, 1999). These tonalite porphyries, which are associated with economically important Cu-Au mineralization and pervasive hydrothermal alteration at Batu Hijau, were emplaced as narrow semi-cylindrical stocks into a broad ENE trending structural dome between ~3.9 – 3.7 Ma (Fig. 2: Garwin, 2000).

Based on petrography and crosscutting relationships they were termed Old Tonalite, Intermediate Tonalite and Young Tonalite (Fig. 3: Meldrum et al., 1994;Clode, 1999).

All three tonalite intrusions are petrographically similar and are geochemically described as low-K calc-alkaline tonalites (Idrus et al., 2007). Least altered specimens contain phenocrysts of plagioclase, hornblende, quartz, biotite, magnetite ± ilmenite hosted in an aplitic groundmass of plagioclase and quartz (Fig. 3: Mitchell et al., 1998;Clode,

1999;Garwin, 2000;Idrus et al., 2007). Notably, all three porphyry intrusions lack potassium feldspar. Identified accessory minerals include apatite, zircon and rare titanites. Relicts of clinopyroxene can be identified within the tonalites. Vein density, ore grade and alteration intensity decrease from the Old to Young Tonalite. The Old Tonalite is the volumetrically smallest occurring mostly at the edges of the composite stock. It can clearly be identified in drill-core where its veins are truncated by later intrusions (Fig. 3) but it is currently not separated from the Intermediate Tonalite by the mine geology

department at Batu Hijau, because their phenocryst proportion is almost indistinguishable. Thus, it is not displayed as a separate unit in Figure 2 but mapped together with the Intermediate Tonalite. It locally contains the highest ore-grades (>1 % Cu and >1 g/t Au) and its matrix is characteristically coarsest of the three tonalite intrusions. The Intermediate Tonalite is the volumetrically largest of the three porphyry intrusions and strongly mineralized (Fig. 2). The Intermediate Tonalite is porphyritic with phenocrysts, including characteristic euhedral quartz phenocryst, <8 mm in diameter (Fig. 3b). The Young

Tonalite is the youngest intrusive rock in the district cutting most vein generations, ore mineralization and alteration (Fig. 3c, e). It is strongly porphyritic with largest observed phenocrysts, including euhedral quartz phenocrysts, and contains elevated but sub-economic metal grades (<0.3 % Cu and <0.5 g/t Au).



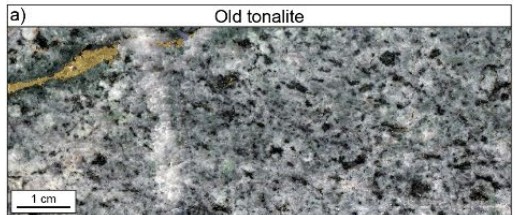

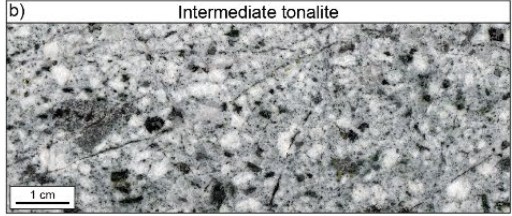

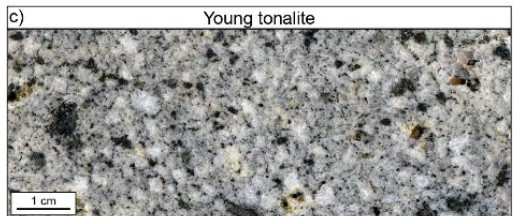

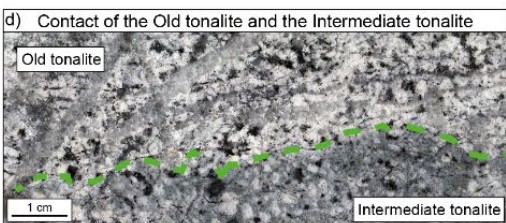

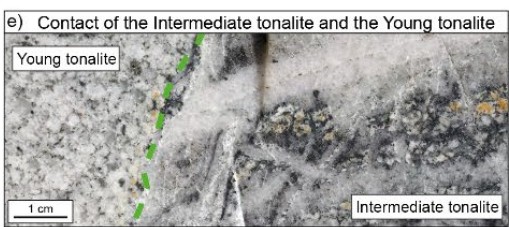

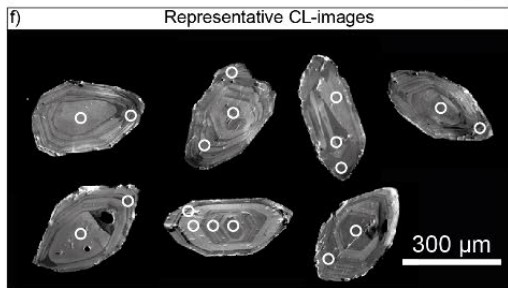





**Figure 3: Rock specimens of the different tonalite porphyries and zircon CL images at Batu Hijau. Mineral assemblage in all tonalites is dominated by plagioclase, quartz and biotite. a) Phenocrysts in the slightly propylitically altered, pre- to syn-Cu-Au-mineralisation, equigranular Old tonalite are <3 mm. b) Phenocrysts of the syn-Cu-Au-mineralisation, porphyritic Intermediate tonalite are < 5 mm. c) The post-mineralisation porphyritic Young tonalite contains largest phenocrysts <8 mm and is characterized by the higher abundance in 'quartz eyes'. d) Abundant veins in the equigranular Old tonalite are truncated by the later porphyritic Intermediate tonalite. e) Strongly veined Intermediate tonalite is truncated by the barren and little altered Young tonalite. Dashed-green lines indicate intrusive contacts. f) Representative zircons that display dominant oscillatory zoning and areas with little zoning. Circles indicate domains selected for LA-ICP-MS analyses (30 μm in diameter).**

Copper and gold are not distributed uniformly within the deposit. High Au-zones are tightly enveloped around the tonalite stocks whereas high copper grades extend further out into the volcanic lithic breccia and the equigranular quartz diorite (Fig. 2). Lowest Cu/Au ratios occur towards and below the current pit floor and higher Cu/Au ratios are recorded peripheral to the central porphyry stock and towards the upper, already mined part of the ore body. A positive correlation between vein density and Cu and Au contents was described at Batu Hijau (Mitchell et al., 1998;Clode, 1999;Arif and Baker, 2004). A-veins were suggested to comprise ~80 % of all quartz veins and contain a similar fraction of the Cu (Mitchell et al., 1998). Most authors suggested that the bulk of the Cu and Au were precipitated as bornite during early A-vein formation and converted to later chalcopyrite and gold associated with AB and B vein formation (Clode, 1999;Arif and Baker, 2004;Proffett, 2009). Other studies on vein relationships and mineralogy using SEM-CP petrography combined with fluid inclusion analyses suggests that Cu-Au ore mineralization including bornite, chalcopyrite and gold all precipitated with a late quartz generation postdating high-temperature A and AB vein quartz, at lower temperature together with the formation of C-veins (Zwyer, 2011;Schirra et al., 2019). Irrespective of the relative timing of stockwork quartz veins and economic ore mineral deposition in the Old and Intermediate tonalites, the Young Tonalite cuts through all high-grade Cu and Au zones demonstrating its late, post-mineralisation emplacement (Fig. 2, 3e). Therefore, the maximum duration of economic mineralisation is bracketed by the emplacement ages of the Cu-Au-rich Old Tonalite pre-dating it and the Young Tonalite post-dating it.

## 3 Materials and Methods

Based on detailed core logging and outcrop mapping with company geologists in May 2016, one sample each from the Old Tonalite, Intermediate Tonalite, the Young Tonalite and the equigranular quartz diorite were selected from locations where the lithologies were in unequivocal time relationship (See Supplementary Material for sample locations). Rocks were crushed and zircons separated with conventional techniques, including Selfrag$^{TM}$ disintegration, panning and heavy liquid mineral separation (methylene iodide; 3.3 g/cm$^3$). Selected zircons were annealed for 48 hours at 900°C, mounted in epoxy



resin and polished to reveal their crystal interior. Polished zircons were carbon coated and imaged using scanning electron microscopy cathodoluminescence (SEM-CL; Tescan EOscan VEGA XLSeries 4 Scanning Electron Microscope) prior to *in situ* LA-ICP-MS analysis for trace elements and U-Pb isotopes employing a 193 nm ASI Resolution (S155) ArF excimer laser with a 30 μm spot diameter, 5Hz repetition rate and 2 J cm$^{-2}$ energy density coupled to an Element SF-ICP-MS. A detailed description of the method including data reduction can be found in Guillong et al. (2014) and the supplementary

material, including results on secondary reference materials. Generally, at least one spot was chosen in the interior (core) and one in the exterior (rim) part of the zircon but up to four individual spots were analysed per zircon (Fig. 3f) to obtain in-situ geochemical information and U-Pb dates. All $^{206}$Pb/$^{238}$U dates were corrected for initial $^{230}$Th-$^{238}$U disequilibrium in the $^{238}$U-$^{206}$Pb decay chain (e.g. Schärer, 1984). Ratios of Th/U recorded by zircons cluster around 0.3 – 0.6 and the dates were therefore corrected assuming a constant Th/U$_{melt}$ of 2 based on partition coefficients (0.25) by Rubatto and Hermann (2007).

Variation of the assumed Th/U$_{melt}$ by ±0.5 would result in changes of individual $^{238}$U-$^{206}$Pb dates of <10 kyr, far below analytical uncertainty.

Titanium concentrations in zircon have been calibrated as a proxy for the crystallization temperature of zircons (Watson and Harrison, 2005;Watson et al., 2006;Ferry and Watson, 2007) and have been widely used in igneous and ore deposit petrology (e.g. Claiborne et al., 2010b;Reid et al., 2011;Chelle-Michou et al., 2014;Dilles et al., 2015;Buret et al.,

2016;Lee et al., 2017). The determination of accurate zircon crystallisation temperatures by Ti-in-zircon thermometry (Ferry and Watson, 2007) requires reliable estimates for the activity of SiO$_2$ and TiO$_2$ (aSiO$_2$ and aTiO$_2$) during zircon crystallization. Based on previous studies on porphyry deposits we utilize an aSiO$_2$ of 1 and an aTiO$_2$ of 0.7 (Chelle-Michou et al., 2014;Buret et al., 2016;Tapster et al., 2016;Lee et al., 2017;Large et al., 2018) reflecting quartz and titanite saturation (Claiborne et al., 2006;Ferry and Watson, 2007). Titanite saturation during zircon crystallization is ambiguous at Batu Hijau

(see discussion) but changes in the assumed aTiO$_2$ result in systematic changes of all zircon crystallization temperatures and will therefore not affect the interpretation of relative temperature changes: a change of the aTiO$_2$ by ±0.2 would result in a variation of about ±30°C.

Imaging by CL and low-precision but spatially resolved LA-ICP-MS U-Pb dates and geochemical data were used to evaluate inherited zircon populations and to select inheritance-free zircons for subsequent dissolution and analysis by high-

precision U-Pb geochronology by CA-ID-TIMS. Selected crystals were removed from the epoxy mount chemically abraded (CA) for 12-15 hours at 180°C using techniques modified from Mattinson (2005). Zircons were spiked with 6-8 μg of the EARTHTIME $^{202}$Pb-$^{205}$Pb-$^{233}$U-$^{235}$U tracer solution (ET2535; Condon et al., 2015;McLean et al., 2015) and dissolved in high-pressure Parr bombs at 210°C for >60 hours. Dissolved samples were dried down and redissolved in 6N HCl at 180°C for 12 hours. Sample dissolution, ion exchange chromatography modified from Krogh (1973) and loading onto zone-refined

Re filaments were conducted at ETH Zürich and are described in detail by Large et al. (2018). High-precision U-Pb isotopic data were obtained employing thermal ionization mass spectrometry at ETH Zürich (Thermo Scientific TRITON Plus). Pb was measured sequentially on a dynamic MassCom secondary electron multiplier and U was measured in static mode as U-oxide using Faraday cups fitted with 10$^{13}$ Ω resistor amplifiers (von Quadt et al., 2016;Wotzlaw et al., 2017).  Data reduction



and age calculation were performed using the algorithms and software described in McLean et al. (2011) and Bowring et al.
(2011). All $^{206}$Pb/$^{238}$U dates were corrected for initial $^{230}$Th-$^{238}$U disequilibrium in the $^{238}$U-$^{206}$Pb decay chain (e.g. Schärer, 1984) using a constant Th/U partition coefficient ratio of 0.25 (Rubatto and Hermann, 2007) assuming that variations in Th/U of the zircons result from different Th/U of the crystallising melt and not from variations in relative zircon-melt partitioning of Th and U. High-precision U-Pb dates were obtained from 45 zircons, all of which were previously analysed by LA-ICPMS.


## 4 Results

### 4.1 Optical zircon appearance and SEM-CL petrography

Zircons were extracted from the three tonalites (Old, Intermediate and Young Tonalite) and the equigranular quartz diorite. Zircon crystals from all three tonalite samples are colourless, euhedral to subhedral and variable in size with c-axis lengths of
100 – 500 µm and aspect ratios between 1:2 and 1:4 (Fig. 3f). Thin section observations reveal zircons that are enclosed by phenocrysts and those that occur within the fine-grained groundmass suggesting protracted zircon crystallisation within the magma until emplacement of the tonalite porphyries. Investigation of mineral separates and mounts with a binocular microscope reveals that many zircons contain small (<<20 µm) mineral or melt inclusions. SEM-CL imaging reveals few unzoned and sector zoned zircon domains, but most zircons exhibit oscillatory zoning (Fig. 3f).
250        Only few broken zircon fragments could be identified from heavy mineral separates of the equigranular quartz diorite, but these indicate originally euhedral to subhedral shapes. Five grains, typically <200 µm long with aspect ratios of ~1:4, could be identified and were mounted. Four zircons were unzoned and one was oscillatory zoned.

### 4.2 Spatially resolved zircon trace element composition

At Batu Hijau, zircon geochemical analyses from the three tonalites display largely overlapping arrays and ranges for all described trace element concentrations and ratios (Fig. 4). Most zircons from the tonalites display systematically higher HREE (e.g. Yb) over MREE (e.g. Dy) and LREE (e.g. Nd) contents in their rims relative to their cores (Fig. 4a). This strongly correlates with core-rim systematics of other differentiation proxies, like increasing Hf or decreasing Th/U (Fig. 4: Hoskin and Ireland, 2000;Claiborne et al., 2006;Schaltegger et al., 2009;Claiborne et al., 2010b;Samperton et al., 2015).
However, some core-rim trends, especially, from the Young Tonalite display increasing Th/U and decreasing Yb/Dy ratios (Fig. 4b).


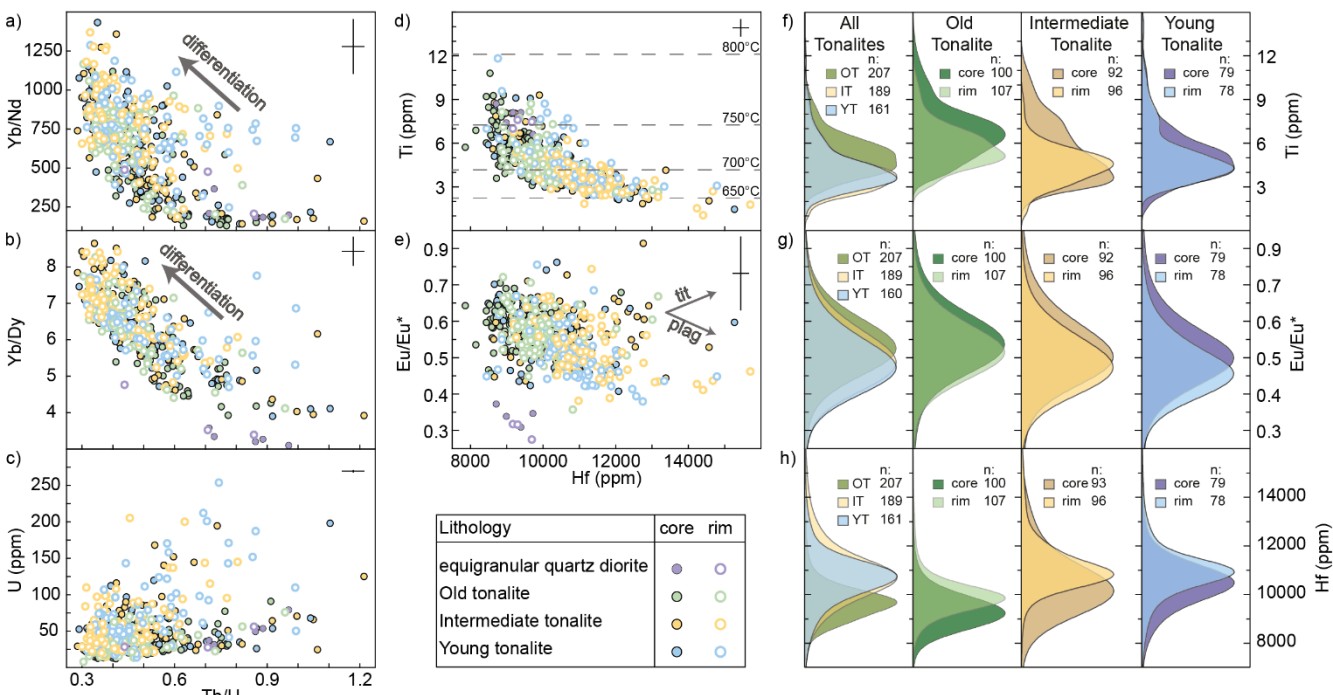

**Figure 4: Covariation diagrams (a-e) and probability density plots (f-h) of in-situ geochemical data obtained by LA-ICP-MS. a) - c) are plotted against Th/U as an indicator for fractionation, whereas d) + e) are plotted against Hf as**

**the fractionation proxy. Arrows labelled 'fractional crystallisation' indicate the approximate predicted direction zircon geochemistry would migrate given fractional crystallization of zircon±apatite ± titanite ± amphibole. Arrows labelled 'plag' and 'tit' points into the predicted direction of zircon geochemistry evolution during co-crystallisation with plagioclase or titanite. Zircons from the three tonalite porphyries are considered to have crystallised from the same magma reservoir, whereas zircons from the equigranular quartz diorite (purple) are unrelated (see text for**

**discussion). Temperature lines in d) are calculated with an assumed aSiO2 = 1 and aTiO2=0.7 based on Ferry and Watson (2007: see text for discussion). Cross in top right corners illustrates average analytical 2σ uncertainties. Probability density plots (after Vermeesch et al., 2013) illustrate differences between different samples and core and rim analyses within each sample. Axes of probability density plots in f) + g) are aligned with axes of d) + e).**


In most zircons, Ti-concentrations decrease from core to rim (Fig. 4d, f). This decrease correlates well with increasing Hf and decreasing Th/U. Maximum and minimum values for all intrusions are ~10 ppm and ~2 ppm resulting in model crystallization temperatures of 770°C to 650°C (see methods for details). The majority of zircons from the Batu Hijau





deposit contain lower U concentrations (<75 ppm) compared to zircons from most other porphyry deposits (several 100
ppm) but individual zircons can contain up to 300 ppm (Fig. 4c). The zircons with high U-concentration do not correspond to
the lower Th/U zircons but also contain high Th-concentrations and cover the whole spectra of Th/U ratios observed at Batu
Hijau (Fig. 4c). The Eu-anomaly (Eu/Eu*, which is a mean to quantify the negative inflexure of the normalised REE
diagram) increases (Eu/Eu* decrease) with increasing Hf concentrations (Fig. 4e). Zircon analyses from the equigranular
quartz diorite plot towards the lowest Hf, Yb/Dy, Yb/Nd, Eu/Eu* highest Th/U and Ti end of the trends displayed by the
tonalite zircons (Fig. 4).

Probability density functions (Vermeesch, 2012) are used to test for statistically significant differences between the
overlapping zircon populations of the different tonalites and between core and rim analyses from the same tonalite
porphyries (Fig. 4f, g, h). The Hf and Ti concentrations as well as the europium anomaly of zircons display overlapping
distributions for the Intermediate and Young tonalites. The Old Tonalite zircon population peaks at higher Ti concentrations
and Eu/Eu* as well as lower Hf concentrations than the younger tonalites. Core and rim analyses from zircons of the Old
Tonalite document decreasing Ti and Eu/Eu* together with increasing Hf concentrations from cores to rims. Hafnium
contents of the rim analyses peak at higher concentrations than the core analyses within the Intermediate and Young Tonalite
with the Eu/Eu* displaying the opposite effect. Populations illustrating titanium concentrations of the two younger tonalites
however, display no systematic changes between core and rim.


## 4.3 CA-ID-TIMS geochronology

We dated 16 zircons each of the Old and Intermediate Tonalite and 13 zircons of the Young Tonalite by high-precision CA-
ID-TIMS geochronology. The youngest zircons of the Old, Intermediate and Young Tonalite yield $^{230}$Th$^{-238}$U disequilibrium
corrected $^{206}$Pb/$^{238}$U zircon dates of 3.736 ± 0.023 Ma, 3.697 ± 0.018 Ma and 3.646 ± 0.022 Ma (Fig. 5). We interpret these
dates as the time of respective porphyry emplacement (c.f. Oberli et al., 2004;von Quadt et al., 2011;Samperton et al.,
2015;Large et al., 2018)) consistent with field observations (Fig. 3). The time intervals between emplacement of the Old and
Intermediate Tonalite and between the Intermediate and Young Tonalite can therefore be constrained to 39 ± 29 ka and 51 ±
28 ka, respectively. Recorded duration of zircon crystallization, as defined by the oldest and youngest zircon of each sample,
spreads over 246 ± 28 kyr, 212 ± 32 kyr and 171 ± 26 kyr for the Old, Intermediate and Young Tonalite (Fig. 5). The overall
310 duration of recorded zircon crystallization is 336 ± 27 ka. Using the youngest zircon population, rather than the youngest
individual zircon as the best approximation for porphyry emplacement (c.f. Samperton et al., 2015;Buret et al., 2016;Tapster
et al., 2016) would result in slightly older emplacement ages (~20 kyr) but nearly identical durations of zircon crystallisation
and time intervals between porphyry emplacement events (see Supplementary Material). Our high-precision CA-ID-TIMS
dates precisely constrain protracted zircon crystallization over several 100 ka and successive emplacement of the three
315 porphyritic tonalite bodies at Batu Hijau within 90 ± 32 ka.



Ratios of Th/U obtained by CA-ID-TIMS analyses on the same sample volume illustrate no systematic variation with time. Values vary inconsistently between 0.4 and 0.6 over the whole recorded time interval (Fig. 6).

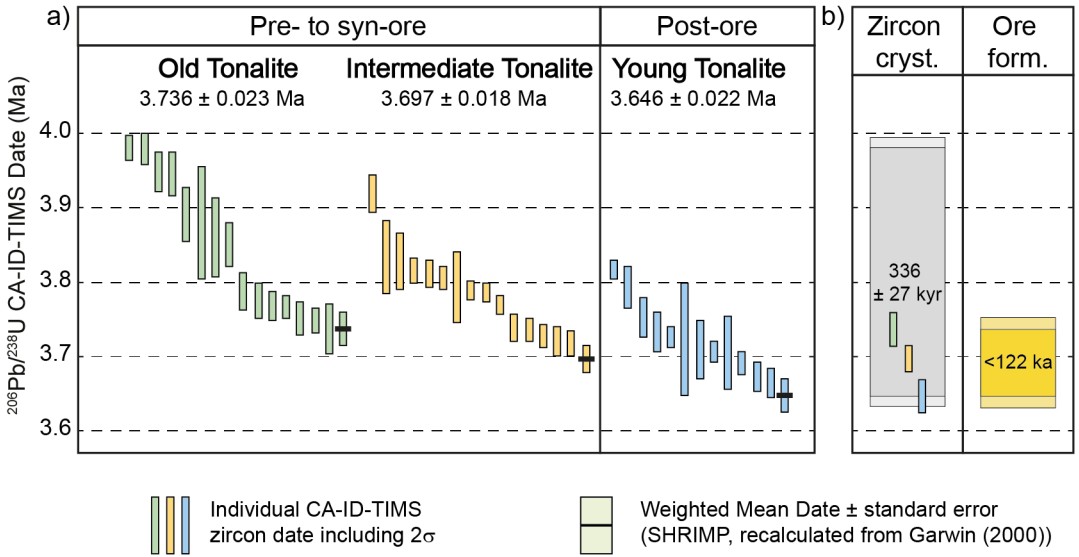

**Figure 5: High-precision U-Pb CA-ID-TIMS zircon dates from the three tonalite porphyries and comparison with weighted mean averages by in-situ U-Pb geochronology. Vertical bars are individual analyses including analytical uncertainty (2σ). The youngest crystallization age is used as the best approximation for porphyry emplacement. The extended range in zircon crystallization ages in each sample indicates protracted crystallization. Yellow box indicates maximum duration of ore formation as constrained by the emplacement age of the Old tonalite and the Young tonalite. Grey box illustrates the duration of zircon crystallization recorded by CA-ID-TIMS geochronology. Vertical bars in the grey box are emplacement ages of the tonalites, demonstrating >200 ka of zircon crystallization before emplacement of the first porphyry intrusion and start of Cu-Au mineralisation.**

**4.4 In-situ U-Pb geochronology**

Trace element and U-Pb isotopic data were obtained for each LA-ICP-MS spot (Fig. 7) prior to CA-IC-TIMS dating. Low Uranium concentrations and the young ages of the analysed zircons resulted in high individual uncertainties for individual in-situ U-Pb dates (Mean: 10%; Minimum: 3%; Maximum: 41%). All individual spot analyses of the three tonalites that were not discarded due to common Pb or strong discordance yield Pliocene dates ($2.98 \pm 1.06 - 4.95 \pm 0.54$ Ma: Fig. 7) with no apparently inherited zircons. All in-situ dates of individual samples illustrate continuous arrays and do not indicate more than one population of zircons per sample (Fig. 7). Weighted means of all zircon analyses from each tonalite are $3.879 \pm 0.027, 0.065, 0.32$ (n = 207, MSWD = 2.1), $3.778 \pm 0.023, 0.061, 0.62$ (n = 189, MSWD = 2.5) and $3.751 \pm 0.023, 0.060,$



0.29 Ma (n = 158, MSWD = 2.6) from oldest to youngest (Fig. 7), where the stated uncertainties are the standard error of the weighted mean, the standard error including an external uncertainty of 1.5 % as suggested by Horstwood et al. (2016) to

incorporate of excess variance, and the standard deviation of zircons dates from each sample. These weighted averages are not overlapping within uncertainty with the emplacement ages constrained by CA-ID-TIMS but overlap with the mean of the respective population. The few in-situ analyses (n = 8) on zircons from the diorite result in overlapping Late Miocene dates. The weighted mean of all LA-ICPMS analyses of the equigranular diorite results in an apparent age of the equigranular diorite of 6.37 ± 0.40, 0.41, 0.37 Ma (n = 8, MSWD = 0.46).


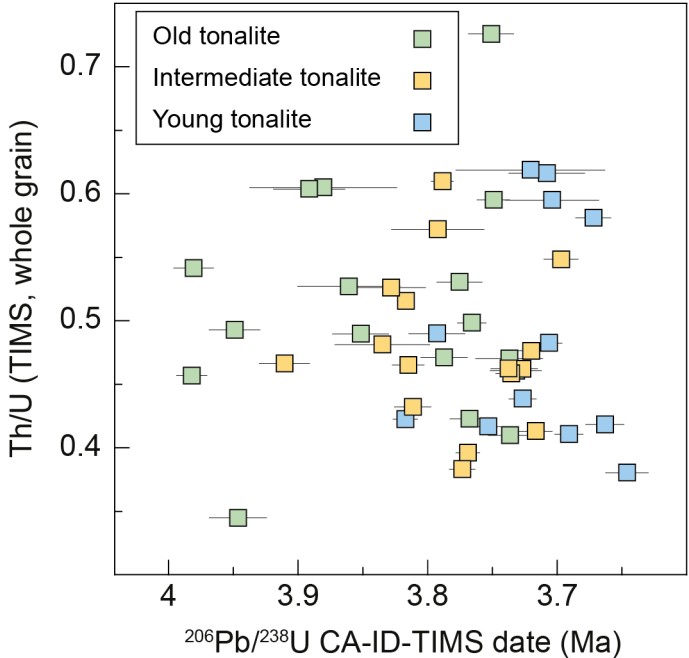

**Figure 6: Th/U ratios plotted against time. Both values obtained from CA-ID-TIMS analyses of the same sample volume.**


Garwin (2000) presented the first SHRIMP U-Pb data on zircons from the Batu Hijau tonalites. Similar to the LA-ICP-MS analyses in this study, individual uncertainties of the dates were elevated (0.12 – 0.30 Ma: ~5 – 10 % uncertainty) due to low U concentrations and the young zircon crystallisation ages. As all dates of each sample appear to represent the same populations (Supplementary Material) weighted means were calculated from of all zircons of a sample, these were

interpreted as the intrusion ages of the tonalites by Garwin (2000). The reported zircon dates were not corrected for [230]Th-





$^{238}$U disequilibrium. For comparability we will only consider zircon dates that are corrected for initial Th/U disequilibrium (Schärer, 1984: for details consult the Supplementary Material). Correction increases individual zircon dates by ~60 – 100 kyr and recalculation of the weighted means averages and standard errors results in dates of 3.74 ± 0.14 Ma (MSWD = 1.2, n = 8), 3.843 ± 0.094 (MSWD = 1.2, n = 18) Ma and 3.81 ± 0.2 Ma (MSWD = 2.35, n = 7) for the Old, Intermediate and

Young tonalite, respectively.

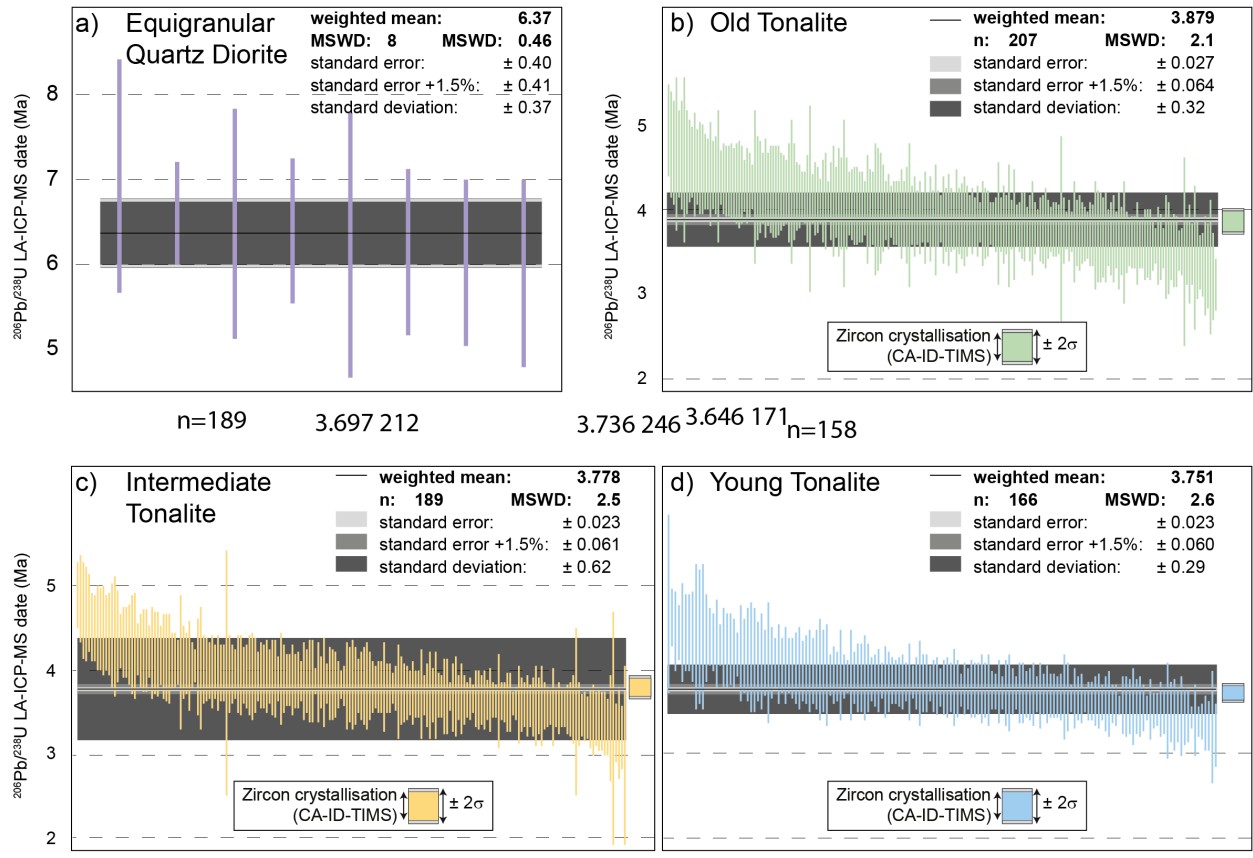

**Figure 7: In-situ U-Pb geochronology by LA-ICP-MS of zircons from the equigranular quartz diorite a), the Old**
**tonalite b), the Intermediate tonalite c) and the Young tonalite d). Vertical lines illustrate individual U-Pb dates including analytical uncertainty (2σ). As no zircon populations can be separated the weighted mean average of all analyses is calculated. Standard error (2SE, lightest grey), standard error + 1.5% (light grey) to incorporate of excess variance (Horstwood et al., 2016) and standard deviation of all individual dates (dark grey) are calculated and plotted. Duration of zircon crystallisation as obtained by CA-ID-TIMS is illustrated as a box for comparison. Lower**



**boundary of the coloured box indicates youngest CA-ID-TIMS date – or the emplacement age. Note the different vertical scale in a). Zircon dates from the equigranular quartz diorite are ~2 Ma older than zircons from the tonalites.**



## 5 Discussion

### 5.1 Timing and duration of magmatic and hydrothermal processes leading to porphyry Cu formation

The three tonalite intrusions each record protracted zircon crystallisation over ~200 kyr, as resolved by high-precision ID-TIMS geochronology. The older zircon dates from the Young and Intermediate tonalites overlap with the younger zircons of the older intrusion/s (Fig. 5). This overlap together with the consistent trace element systematics of the three samples (Fig. 4) strongly suggests crystallisation of all zircons within the same magma reservoir. High-precision geochronology records a total duration of zircon crystallisation of $336 \pm 27$ kyr, which is also a minimum estimate for the lifetime of the deeper reservoir underlying Batu Hijau. The first exposed and highly mineralised tonalite intrusion (Old Tonalite) was injected into the upper crust $246 \pm 28$ kyr after the onset of zircon crystallization. Emplacement of the three tonalites occurred within $90 \pm 32$ kyr. Emplacement of the Old Tonalite was followed by the emplacement of the Intermediate Tonalite after $39 \pm 29$ kyr and the Young Tonalite was emplaced after a further $51 \pm 28$ kyr.

The maximum duration of ore formation is defined by the timespan between the emplacement of the pre- to syn-mineralisation Old Tonalite and the post-mineralisation Young Tonalite (Fig. 3d, e) and can be therefore constrained to less than 122 kyr. This maximum duration is in good agreement with previous geochronological studies indicating timescales of ore formation from <100 kyr to <29 kyr (Fig.8, 9: von Quadt et al., 2011;Buret et al., 2016;Tapster et al., 2016). It is also coherent with results from thermal modelling studies (Cathles, 1977;Weis et al., 2012) and modelling of diffusive fluid-rock equilibration (Cathles and Shannon, 2007;Mercer et al., 2015;Cernuschi et al., 2018) suggesting timescales of ore formation between a few ka and 100 kyr. Strongly elevated Cu- and Au-grades in the Old Tonalite and somewhat lower, but still economic, grades within the Intermediate Tonalite (Clode, 1999;Garwin, 2000;Arif and Baker, 2004) together with cross-cutting relationships (Fig. 3d, e) indicate that mineralisation occurred within at least two but possibly more pulses: (i) one strong mineralisation pulse associated with or slightly postdating the emplacement of the Old Tonalite but predating the injection of the Intermediate Tonalite (Fig. 3d); (ii) a second pulse is bracketed by the intrusion of the Intermediate and the Young Tonalite (Fig. 3e). More than one episode of mineralisation is also inferred based on detailed mineralogy and vein petrography (see Geology section: Arif and Baker, 2004;Zwyer, 2011). This further strengthens the hypothesis that individual ore-forming hydrothermal pulses are relatively short events, possibly on the millennial or sub-millennial scale (Cathles, 1977;Weis et al., 2012;Mercer et al., 2015), but that the formation of large economic Cu-Au deposits occurs in several pulses occurring over a few 10s of kyr but ≤100 kyr (von Quadt et al., 2011;Weis et al., 2012;Cernuschi et al., 2018).





**5.2 Reconstructing the chemical and physical evolution of a porphyry-forming magma reservoir**

Trace element systematics of zircons are powerful geochemical proxies, if applied correctly, as they record the magma
evolution and characterise the magmatic system, that they crystallised from (e.g. Hoskin and Schaltegger, 2003;Reid et al.,
2011;Schoene et al., 2012;Wotzlaw et al., 2013;Chamberlain et al., 2014;Samperton et al., 2015). The largely overlapping
trace element systematics recorded by zircons together with the protracted nature of zircon crystallisation are here used to
infer zircon crystallisation within the same mid- to upper crustal magma reservoir that sourced magmas forming the three
tonalitic porphyry stocks but most likely also volatiles and metals to form the porphyry Cu-Au deposit. At Batu Hijau we are
able to reconstruct the magmatic evolution over $336 \pm 27$ kyr of recorded zircon crystallization.

Th/U ratios and Hf concentrations are commonly used as proxies for the degree of crystal fractionation within a
magma reservoir (e.g. Claiborne et al., 2006;Claiborne et al., 2010b;Samperton et al., 2015). The systematically decreasing
Th/U ratios and increasing Hf concentrations between samples and from cores to rims (Fig. 4) are indicative of progressive
melt differentiation during zircon crystallisation. The good correlation of these melt evolution proxies with decreasing Ti-
contents (Fig. 4) further suggests progressive cooling during differentiation. Ratios of HREE over MREE or LREE (e.g.
Yb/Dy, Yb/Nd) can be utilised to make inferences about the co-crystallising mineral assemblage. Titanite for example
preferentially depletes the melt in MREE resulting in distinct trace element patterns recorded by co-crystallizing zircon (e.g.
Reid et al., 2011;Wotzlaw et al., 2013;Samperton et al., 2015;Loader et al., 2017). The systematically higher HREE (e.g. Yb)
over MREE (e.g. Dy) and LREE (e.g. Nd) contents in the rims of most zircons relative to their cores (Fig. 4a) thus indicate
zircon crystallisation from a fractionally crystallising magma with co-crystallisation of minerals that preferentially
incorporate MREE and LREE (e.g. apatite, titanite, amphibole). At Batu Hijau apatites were petrographically identified,
whereas magmatic titanite occurs very subordinately. The apparent lack of magmatic titanite is unusual as it is reported as a
common accessory phase in many other porphyry-Cu deposits (e.g. Bajo de la Alumbrera, El Salvador, Ok Tedi, Oyu
Tolgoi). The absence of euhedral titanite within the mineral separates could be the result of dissolution during intense
hydrothermal alteration (van Dongen et al., 2010). The decrease of Eu/Eu* correlating with proxies of increased
fractionation (Hf, Fig. 4e)  and during zircon growth (Fig. 4f) suggests co-crystallisation of plagioclase and could indicate a
lack of titanite crystallisation, or subordinate crystallisation, as already minor titanite crystallisation strongly increases the
Eu/Eu* recorded by zircon (Loader et al., 2017). This apparent lack of titanite crystallisation identify apatite as the main
REE fractionating mineral during zircon crystallization.
Trace element compositions of zircons from the equigranular quartz diorite suggest crystallisation within a hotter
and less evolved magma than the zircons from the tonalites (Fig. 4). In principle, this might indicate that all zircons analysed
in this study have crystallised from the same reservoir. However, the >2 Ma time gap is longer than the thermal lifetime of
any recognised upper-crustal magmatic body (e.g. Schoene et al., 2012;Wotzlaw et al., 2013;Caricchi et al., 2014;Samperton
et al., 2015;Eddy et al., 2016;Karakas et al., 2017) and longer than considered possible based on thermal modelling (Jaeger,





1957;Annen, 2009;Barboni et al., 2015). We therefore consider the zircons within the equigranular Diorite to be part of a separate crustal magmatic system not directly related to the ore-forming system that sourced the three tonalitic intrusions.

Trace element populations of zircons from the three tonalites demonstrate that the crystallising magma at the time of emplacement of the Old Tonalite was hotter and less fractionated (Fig. 4) than at the time of emplacement of the younger Intermediate and Young Tonalite (i.e. 39 ± 29 ka and 90 ± 32 ka after emplacement of the Old Tonalite, respectively). The

good correlation of proxies indicating progressive differentiation (Th/U and Hf) with decreasing Ti concentrations (Fig. 4d) indicates that the magma reservoir cooled during concurrent crystallisation and melt evolution. In-situ analyses of cores and rims are evidence for an evolving magma reservoir over the course of individual zircon crystallisation (decreasing Hf: Fig. 4h). Core-rim systematics of zircons from the Old Tonalite further demonstrate cooling during protracted zircon growth (Fig. 4f). Rarely recorded coherent zircon trace element systematics recording melt differentiation over time are commonly

inferred to result from zircon crystallisation within a homogeneous magma that best resembles near-closed-system behaviour (e.g. Wotzlaw et al., 2013;Large et al., 2018). The lack of such systematic temporal changes in the chemistry of the zircons (Fig. 6) indicates that the magma reservoir at Batu Hijau was not evolving homogenously. This could be explained by incremental recharge or assembly of the magma reservoir. However, this would imply at least partial resetting of the intra-grain systematics recorded in zircons from the Old Tonalite (Buret et al., 2016;Large et al., 2018). To explain the intra-grain

and inter-sample systematics but absence of temporal trends (Fig. 4, 6), we favour different degrees of crystallinity in the magma reservoir. Overall the reservoir is generally hotter and less evolved at the time of emplacement of the Old Tonalite than thereafter (Fig. 4). We therefore suggest that the magma reservoir underlying Batu Hijau progressively but heterogeneously cooled and crystallised over at least 246 ± 28 ka with potential incremental recharges until emplacement of the Old Tonalite.

A change from a differentiating, crystallising and cooling magma reservoir to a state of chemical and thermal stability is recorded between emplacement of the Old and Young Tonalite (separated by 90 ± 32 kyr) as demonstrated by the trace element systematics of the Intermediate and Young Tonalite porphyries. The indistinguishable highly fractionated and low temperature zircon characteristics (Fig. 4) indicate that the magma reservoir remained in near steady-state conditions between emplacement of the Old and Young Tonalite as coherent intra-grain systematics are not pronounced (Hf) or absent

(Ti) in zircons from the younger tonalites (Fig. 4f, h).

Irregular zircon trace element systematics in other intrusive magmatic settings have been associated with crystallisation in non-homogenised and small melt batches sometimes with contemporaneous incremental magma addition to the mushy magma reservoir (e.g. Schoene et al., 2012;Buret et al., 2016;Tapster et al., 2016). Geochemically similar zircon chemistries of the Intermediate and Young Tonalite could also result from chemical stability as the magma reservoir reached

the 'petrological trap' at a crystallinity of ~55 – 65% (Caricchi and Blundy, 2015) where the crystal fraction does not change over a broad temperature interval. Rim analyses that plot outside the mineral co-crystallisation trends than the respective core analyses (Fig. 4) could suggest late-stage crystallisation within a nearly solidified magma that can be characterized by unsystematically variable trace element systematics (Buret et al., 2016;Lee et al., 2017). Alternatively, they could indicate





thermal and possibly chemical rejuvenation of the magma (Buret et al., 2016). The latter would help explaining the recorded thermal stability over several 10s of kyr. It is not possible to unambiguously identify one of the two mechanisms as dominant and a concurrence of both is feasible. We therefore propose that in between emplacement of the Old and Young Tonalite the underlying magma reservoir was in a thermally and chemically stable and crystal-rich state and was most likely affected by incremental magma recharge or underplating.

Our data of a porphyry-Cu fertile magmatic system constrain a heterogeneous magma reservoir that was initially
dominated by cooling and melt differentiation and evolved into a thermally and chemically stable, crystal-rich magma that possibly experienced incremental recharge. The likely transitional change of reservoir behaviour can be temporally constrained to have occurred between emplacement of the Old and Young tonalites and coincides with the formation of a world-class Cu-Au reserve. This suggests that porphyry Cu-Au deposits form after a few 100 ka of cooling and crystallisation, potentially within an originally melt-rich magma reservoir.

**5.3 Different timescales of processes related to porphyry Cu-Au ore-formation**

To date no clear relationship between the duration of magmatic-hydrothermal activity and the size of porphyry deposits can be identified from studies applying high-precision CA-ID-TIMS geochronology. Comparison of published datasets (Buret et al., 2016;Tapster et al., 2016;Large et al., 2018) reveals maximum durations of metal forming events between a few $10^4$ to $10^5$ yr (Fig. 9). Although these studies are so far constrained to deposits of <10 Mt contained Cu they range over at least one
order of magnitude in size (Koloula vs. Batu Hijau). A correlation between the duration of the mineralizing/magmatic event and the total mass of deposited copper had been previously suggested based on compilations of different geochronological data-sets (Chelle-Michou et al., 2017;Chiaradia and Caricchi, 2017;Chelle-Michou and Schaltegger, 2018;Chiaradia, 2020). High durations of ore formation (>1 Ma) were suggested based on Re-Os geochronology on Molybdenite at the giant porphyry deposits and deposit clusters in Chile (>50 Gt Cu: El Teniente, Cannell et al. (2005) and Maksaev et al. (2004); Rio
Blanco, Deckart et al. (2012); and Chuquicamata, Barra et al. (2013)). Copper (-gold) mineralising timescales were calculated by subtracting the youngest from the oldest Re-Os date. However, recent Re-Os dates from El Teniente (Spencer et al., 2015) indicate that the spread in dates is more consistent with several short (≤200 kyr) hydrothermal events separated by hiatuses of ~500 kyr. Thus, the large tonnage of these deposits could be the result of the superimposition of several ore forming mid- to upper crustal magmatic systems. As the correlation of deposit size and timescales of shallow magmatic-
hydrothermal systems is currently ambiguous we would argue that other variables could be the dominant factors controlling the deposit size, such as magma reservoir size, magma or fluid chemistry, fluid release and focussing mechanisms or the metal precipitation efficiency.

Zircon crystallisation over ~200 kyr before the onset of porphyry-ore formation recorded at Batu Hijau is consistent with other high-precision geochronological studies on porphyry deposits (Fig. 8, 9: Buret et al., 2016;Tapster et al.,
2016;Large et al., 2018). The lack of variation observed in these deposits suggests the necessity of a long-lived and continuously crystallising magma reservoir preceding economic ore formation. The recorded ~200 kyr of protracted zircon





crystallisation could indicate a period of volatile enrichment as a result of fractional crystallisation and cooling of the magma reservoir before porphyry emplacement.

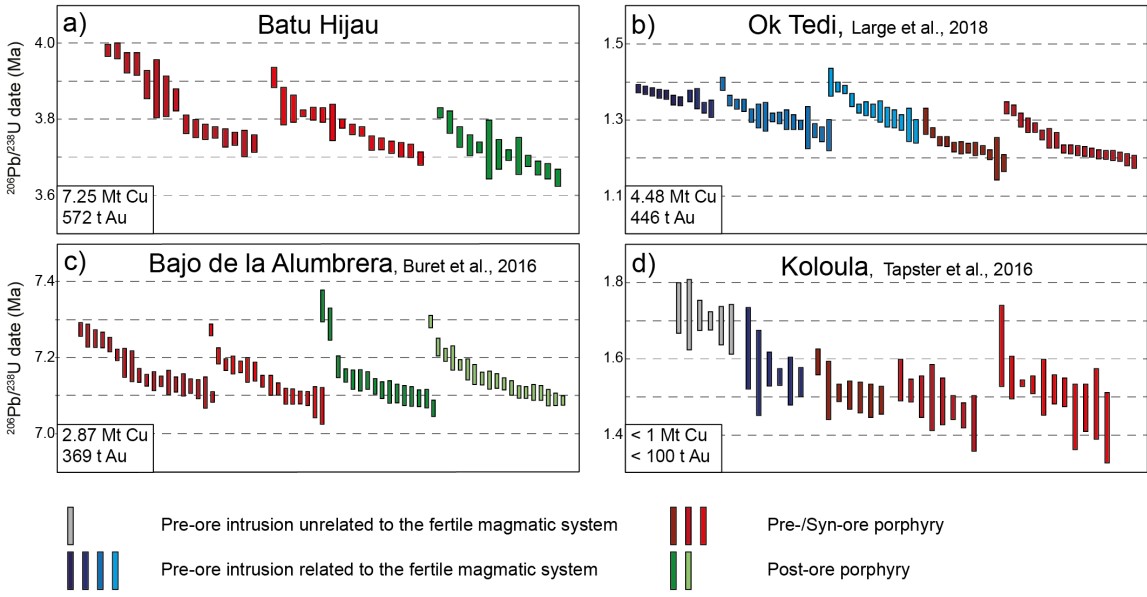


**Figure 8: Compilation of high-precision data-sets on several pre-, syn- and post-ore intrusions at magmatic-hydrothermal Cu-Au deposits. Data for Ok Tedi, Bajo de la Alumbrera and Koloula are from Large et al. (2018) Buret et al. (2016) and Tapster et al. (2016), respectively. Coloured vertical bars are individual analyses including analytical uncertainty (2σ). Intrusions are categorised in pre-ore, pre-/syn-ore and post-ore intrusion. Decreasing**
**deposit size from left to right (tonnages from Cooke et al., 2005).**

The geochronological data from the Batu Hijau district are further evidence that rapid porphyry emplacement and ore formation (<100 ka) are the product of a longer term evolution (a few 100 ka) of a large magma reservoir underlying the
porphyry deposit that is the main driver of ore formation (von Quadt et al., 2011;Chelle-Michou et al., 2014;Buret et al., 2016;Tapster et al., 2016;Buret et al., 2017;Large et al., 2018). Magma reservoirs capable of forming porphyry deposits are in turn part of a longer-term (several Myr) evolution of lithosphere-scale magma systems (Sasso, 1998;Rohrlach et al., 2005;Longo et al., 2010;Rezeau et al., 2016), which is consistent with the >>2 Myr record of intrusive rocks preceding porphyry emplacement and ore formation recorded in the Batu Hijau district (Garwin, 2000;Wawryk and Foden, 2017).




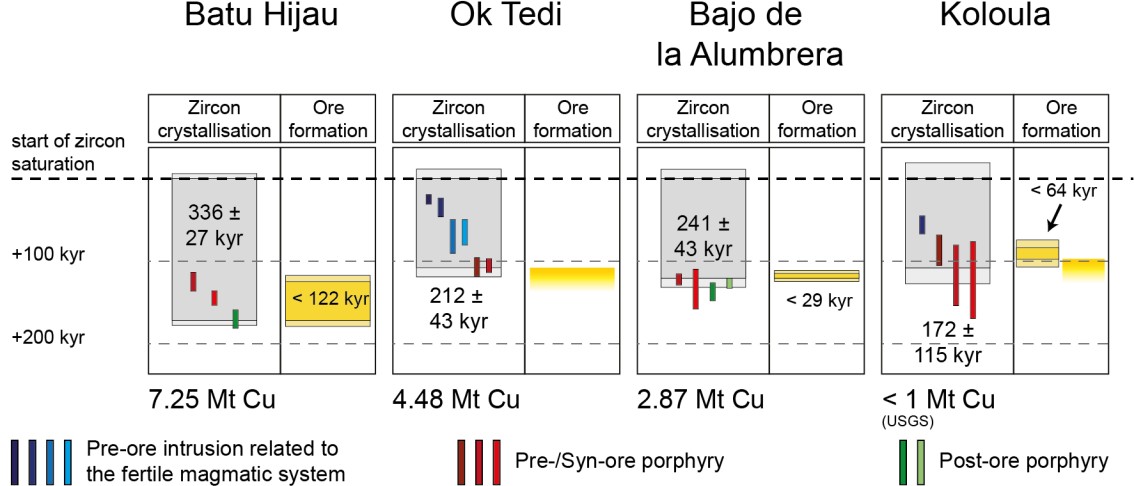

**Figure 9: Overview of high-precision geochronology studies on porphyry deposits.**
**Data for Ok Tedi, Bajo de la Alumbrera and Koloula are from Large et al. (2018) Buret et al. (2016) and Tapster et al. (2016), respectively. Coloured vertical bars are emplacement ages of different intrusive rocks described from these deposits. Intrusions are categorised in pre-ore, pre-/syn-ore and post-ore intrusion. Decreasing deposit size from left to right (tonnages from Cooke et al., 2005). Diagrams are aligned that the onset of zircon crystallisation overlaps in all deposits. Grey bars indicate recorded duration of zircon crystallisation. Yellow bars illustrate maximum durations of total ore formation or individual ore formation pulses. Yellow bar fading out downwards indicates the absence of a post-ore intrusion and the inability to constrain total duration of ore formation. Note that we excluded the sample X176 from Koloula as it is not related to magmatic history leading to ore formation (Tapster et al., 2016).**

**5.4 Resolving lower crustal magmatic processes from Zircon petrochronology**

The lack of inheritance within the zircon record at Batu Hijau suggests that the crustal magmas experienced very minor crustal assimilation. Typically, magmas that are associated with porphyry ore formation contain diverse suites of inherited zircons (e.g. Tapster et al., 2016;Lee et al., 2017;Large et al., 2018), which have been interpreted to represent extended interaction with arc lithologies (Miller et al., 2007). This apparent lack of crustal contamination is consistent with the juvenile isotopic signatures (Pb-Pb, Sm-Nd, Rb-Sr) of intrusions in the Batu Hijau district (Garwin, 2000;Fiorentini and Garwin, 2010). The juvenile and "porphyry-fertile" magmas at Batu Hijau have been explained by asthenospheric mantle upwelling through a tear in the subducting slab that resulted from the collision with the Roo rise (Garwin, 2000;Fiorentini and Garwin, 2010). This would also explain why the only mined porphyry-deposit in the Sunda-Banda arc (Batu Hijau) and the most promising prospects (Elang and Tumpangpitu) are located above the inferred margin of the subducting Roo rise (Fig. 1).

The formation of porphyry Cu(-Au) deposits has been commonly associated with the fractionation of amphibole ± garnet in thickened crust (e.g.Rohrlach et al., 2005;Lee and Tang, 2020) within lower crustal magma reservoirs that are



active over several Myr (Rohrlach et al., 2005). Zircons have been suggested to directly track this extended lower crustal
       history (Rohrlach et al., 2005). At Batu Hijau no zircon was identified that crystallised resolvably before the main
       crystallisation period, which we consider to have occurred in the mid- to upper crust (Fig. 5, and discussion above). Unzoned
       cores surrounded by oscillatory zoned rims (Fig. 3f) could be interpreted to reflect a two-stage crystallisation process
       however, the depth of these two processes cannot be resolved and they would have occurred within the few 100 kyr of
recorded zircon crystallisation (Fig. 5). As most crystals within a mount are not polished exactly to their centre, the unzoned
       cores could equally likely represent a polishing effect where the surface of one zone appears as an unzoned core. Therefore,
       it is highly speculative to directly relate zircon textures to a locus or style of zircon crystallisation.

               In the case of Batu Hijau, petrochronology data was used to reconstruct the mid- to upper crustal magma evolution
       but the data can only provide indirect information about the lower crustal processes involved in the formation of the deposit.
For example, the overall elevated Eu/Eu* of the investigated zircons (0.4 – 0.7; cf. Loader et al., 2017) could be the result of
       amphibole fractionation in the lower crust, which would have, relatively, enriched the residual melt Eu compared to the other
       REE. This would be analogous to elevated whole-rock Sr/Y ratios in exposed rocks being indicative of the lower crustal
       fractionating assemblage (Rohrlach et al., 2005;Chiaradia, 2015). The intra-crystal and intra-sample trends of decreasing
       Eu/Eu* discussed above describe the evolution within the mid- to upper crustal magma reservoir that was dominated by
plagioclase crystallisation and do not reflect any lower crustal process. Zircon can thus directly record the mid- to upper
       crustal magma evolution but the information about lower crustal processes is limited to potentially identifying the chemistry
       of melt and magma that was injected from below into the mid- to upper crust, where zircon started crystallising.

**5.5 An assessment of the accuracy and precision of CA-ID-TIMS and in-situ U-Pb zircon geochronology**

       The obtained U-Pb dataset from Batu Hijau, allows a critical comparison of the two zircon U-Pb geochronology techniques
(LA-ICP-MS, CA-ID-TIMS) that have different analytical precision and can analyse samples on varying spatial scales.
       Previous investigation of the same lithologies by SHRIMP (Garwin, 2000) allows further comparison. The spatially resolved
       and fast in-situ U-Pb geochronology techniques (LA-ICP-MS or SIMS/SHRIMP) allow the investigation of different crystal
       domains, whereas the much more time-consuming CA-ID-TIMS analysis of zircons or zircon fragments provides the highest
       analytical precision. The in-situ techniques can discriminate between different zircon populations within single crystals (e.g.,
inheritance), whereas CA-ID-TIMS geochronology allows for an >10-fold analytical precision for individual grains that is
       required to resolve rapid geochronological events. To increase precision of the in-situ techniques large numbers of individual
       dates that are considered to represent the same geological event are commonly used to calculate a weighted mean date and
       standard error of the mean (Wendt and Carl, 1991). On the other hand, the CA-ID-TIMS community has started to measure
       only small zircon fragments to increase spatial resolution (e.g. Samperton et al., 2015;Smith et al., 2019)). Here, the
comparison of the different U-Pb zircon techniques applied to the same rock suite allows an assessment of the accuracy of
       the techniques and of the effect of statistical treatment on the accuracy and precision of the different techniques.



At Batu Hijau, the youngest individual CA-ID-TIMS U-Pb date of each sample is used as the best approximation for the emplacement age of the respective porphyry. This is based on the assumption that the magma cooled rapidly upon injection into the subvolcanic environment (cf. Schaltegger et al., 2009;von Quadt et al., 2011;Samperton et al., 2015;Large et al., 2018). The resulting porphyry emplacement ages are 3.736 ± 0.023 Ma, 3.697 ± 0.018 Ma, 3.646 ± 0.022 Ma for the Old, Intermediate and Young Tonalite, respectively (Fig. 5). The extended range of concordant zircon dates obtained by CA-ID-TIMS does not allow to distinguish between different stages of zircon crystallisation within each sample (e.g., inherited vs. autocrystic) but the common geochemical trends indicate crystallisation within the same magma reservoir (see above). Thus, the range in zircon dates preceding this emplacement age is interpreted to represent zircon crystallisation within the underlying source magma reservoir over parts or, depending on the timing of onset of zircon saturation, the entirety of its lifetime. The recorded duration of zircon crystallisation is 336 ± 27 kyr.

Similar to the CA-ID-TIMS dates, in-situ analyses by LA-ICP-MS illustrate an extended range of zircon dates that cannot be separated into different stages of zircon crystallisation. However, the span in zircon dates is about a magnitude higher for the in-situ analyses (1.41 ± 0.5 – 2.1 ± 1.1 Myr) than obtained by CA-ID-TIMS (0.171 ± 0.026 – 0.246 ± 0.028 Myr). This could indicate that the LA-ICP-MS data records an extended period of zircon crystallisation not covered by CA-ID-TIMS data, potentially due to sampling bias, that one data-set is inaccurate or that the span within the in-situ data is the result of analytical scatter.

Sampling bias in the selection of the zircons for CA-ID-TIMS geochronology can be excluded as the analyses were conducted on chemically abraded zircons (Mattinson, 2005) that cover the oldest and youngest dates obtained by LA-ICP-MS (Fig. 10). High accuracy of both, the CA-ID-TIMS and LA-ICP-MS, data-sets are suggested by routine measurements of secondary standards during the LA-ICP-MS analytical run (See Supplementary Material) and regular measurements of zircon standards by CA-ID-TIMS over the period of data acquisition (von Quadt et al., 2016;Wotzlaw et al., 2017). The distributions of the zircon dates of each sample, as illustrated by probability density plots (Fig. 11), illustrate that the peak of the LA-ICP-MS and SHRIMP dates falls within the mean of zircon crystallisation as defined by the CA-ID-TIMS data-set. This suggests that all datasets are accurate but that the in-situ data displays more scatter and lower precision. LA-ICP-MS analyses record younger zircon dates for core analyses than rim analyses in 17 of 49 cases, however the dates are always overlapping within uncertainty. Direct comparison of U-Pb dates from the same zircon crystals by the two techniques (Fig. 10) reveals that suggested dates from the two techniques do not overlap within uncertainty in some cases (6/49 for rim analyses). This could indicate that uncertainties associated with the LA-ICP-MS data have been underestimated in relation to the achieved precision of the technique. However, due to the high number of analyses it is more likely that it is purely an effect of analytical scatter where 5% of the data do not fall within the 95% confidence interval. This is corroborated by ~7 % (39/554) of LA-ICP-MS dates not overlapping with the minimum overall duration of zircon crystallisation identified by CA-ID-TIMS dates from all porphyries (336 ± 27 kyr). It is therefore concluded that all three techniques are accurate and represent the ~300 – 350 kyr of zircon crystallisation. The high number of analyses obtained by LA-ICP-MS together with





610  the lower precision results in extreme outliers that extend the apparent duration of zircon crystallisation but can be regarded

purely as an analytical artefact.

**Figure 10: Comparison of in-situ LA-ICP-MS dates and CA-ID-TIMS dates on the same zircons. In upper panel each**
615  **CA-ID-TIMS date is aligned with the rim (filled) and core (empty) LA-ICP-MS date of the same zircon. Coloured**
**bars indicate individual CA-ID-TIMS analysis including analytical uncertainty (2σ). Downward pointing black arrow**
**indicates that core analyses are older than rim analyses of the respective zircon, wherease red upward pointing arrow**
**indicates the opposite. Note that CA-ID-TIMS dates can be plotted several times, with core and rim analyses of the**
**same zircon. Lower panel compares CA-ID-TIMS date with respective rim analysis.**

620




## 5.6 Determining geological ages, uncertainties and rates from in-situ U-Pb data

Understanding the timing of magma emplacement, crystallisation or eruption is essential for determining dates and rates of
magmatic processes and those directly related or bracketed by them. Where high-precision CA-ID-TIMS data is not
available porphyry emplacement ages are commonly inferred by calculating a weighted mean and standard error from the
youngest overlapping population of in-situ U-Pb dates (e.g. Correa et al., 2016;Rezeau et al., 2016;Lee et al., 2017). In the
case of Batu Hijau the calculation would include all LA-ICP-MS zircon dates for each sample as there is no apparent
inheritance within the datasets (Fig. 7). The resulting weighted mean dates for the Old, Intermediate and Young Tonalite are
3.879 ± 0.027/0.064 (MSWD = 2.1, n = 207), 3.783 ± 0.023/0.061 (MSWD = 2.5, n = 189 ), 3.751 ± 0.023/0.060 (MSWD =
2.6, n = 158) where the first stated uncertainty is the standard error including internal uncertainties and those associated with
tracer calibration (Schoene, 2014) and the second includes the added 1.5% external uncertainty as suggested by Horstwood
et al. (2016) to account for excess variance. The MSWD for each data-set (2.1 – 2.6) is elevated in respect to the sample size
(n=150-200; Wendt and Carl, 1991) suggesting an underestimation of the individual uncertainties or that the data do not
represent a normal distribution, e.g. by prolonged zircon crystallisation. However, there is no obvious treatment of the data
to obtain more appropriate MSWDs. Under these conditions weighted means and standard errors should not be calculated
(Wendt and Carl, 1991) but we will ignore this here, as is commonly done in the scientific literature, and will use these
numbers to illustrate a few points below. Analogous, the weighted mean and standard error of all zircons analysed by
SHRIMP from each sample results in weighted means of 3.74 ± 0.14 Ma (MSWD = 1.2, n = 8), 3.843 ± 0.094 (MSWD =
1.2, n = 18) Ma and 3.81 ± 0.2 Ma (MSWD = 2.35, n = 7) for the Old, Intermediate and Young tonalite, respectively (Fig.
11). The weighted means of the different tonalites obtained by LA-ICP-MS would be in accordance with cross-cutting
relationships, whereas the SHRIMP dates overlap within uncertainty. The calculated standard errors for the LA-ICP-MS
dates are significantly smaller than for the SHRIMP data. The decrease in the standard errors is directly correlated with the
increasing sample size (Wendt and Carl, 1991;McLean et al., 2011b) implying that a comparably high number of SHRIMP
analyses would result in similarly low standard errors. Irrespective of the different standard errors the calculated weighted
means by SHRIMP and LA-ICP-MS are overlapping within uncertainty, thus suggesting that both are accurate or similarly
inaccurate.

At Batu Hijau, emplacement ages determined by CA-ID-TIMS geochronology are systematically younger than the
weighted mean dates calculated from in-situ data (100 – 150 kyr: except CA-ID-TIMS and SHRIMP for the Old Tonalite)
and the ages determined by CA-ID-TIMS do not overlap with the LA-ICP-MS values within the attributed uncertainties (Fig.
11). Indeed, disparities between different U-Pb data-sets on the same porphyry samples have been noted in several studies
comparing high-precision CA-ID-TIMS data with in-situ data (von Quadt et al., 2011;Chiaradia et al., 2013;Chelle-Michou
et al., 2014;Chiaradia et al., 2014;Correa et al., 2016). As discussed above all presented datasets are considered accurate and
thus the discrepancy in dates is most likely the result of differences in the statistical handling and geological interpretation.




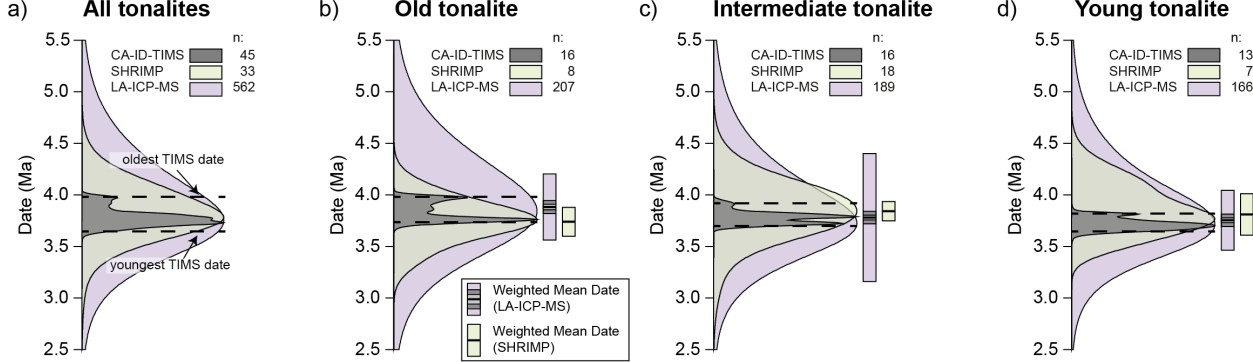

**Figure 11: Probability density plots for the geochronology data for each analytical technique. All dates of each**
**technique are combined in a). Plots in b), c), and d) are constructed from the data of the Old, Intermediate and**
**Young tonalite. Dashed lines indicate the youngest and oldest zircon crystallization age as determined by CA-ID-**
**TIMS for the respective investigated data-set. Weighted means, standard error, standard error + 1.5% (Horstwood et**
**al., 2016) and standard deviation are identical to those in Fig. 5.**


        The protracted zircon crystallisation identified at Batu Hijau has profound implications for the determination of

magma emplacement, crystallisation or eruption ages. Extended magma reservoir lifetimes are not unique to Batu Hijau but

a commonly described feature (e.g. Miller et al., 2007;Claiborne et al., 2010a;Reid et al., 2011;Buret et al., 2016). A

weighted mean is a measure to quantify the mean of a population whereby emphasising the importance of values with low

uncertainties over those with high uncertainties (Reiners et al., 2017) and is only allowed to be used in cases where the data

is normally distributed around the expected value (Wendt and Carl, 1991). The presented in-situ data-sets record protracted

zircon crystallisation (>300 kyr ) in the magma reservoir that results in zircon population distributions that cannot be easily

defined statistically (Fig. 5, 7: cf. Keller et al., 2018), negating a normal distribution around a single geological event – the

emplacement age. The calculated weighted mean rather represents the mean of the duration of zircon crystallisation,

corroborated by the weighted means of the LA-ICP-MS and SHRIMP dates approximately describing the mean of the zircon

populations defined by CA-ID-TIMS (Fig. 11). Therefore, the calculated weighted mean does not describe any specific

geological event, especially as the uncertainties indicated by the standard error for the LA-ICP-MS data are too small to even

cover the entire recorded duration of zircon crystallisation. It has to be noted that more complex settings where xenocrystic

zircons overlap within uncertainty of the in-situ techniques with the auto-and antecrystic zircon population (e.g. Chelle-

Michou et al., 2014) would result in even less reliable geological dates estimated by in-situ techniques. Therefore, using the

weighted mean of a zircon population to determine emplacement or intrusion ages can be a broad oversimplification if dates




do not represent a normal distribution around the dated event. Similar problems can occur when calculating eruption ages for non-homogeneous zircon populations from tuffs or other volcanic rocks (Schoene, 2014).

Traditionally, problems associated with the oversimplification associated with calculating weighted means and their standard errors in geochronology were hidden by the higher uncertainties resulting from higher analytical uncertainties and smaller sample sizes. The standard error of the mean is a measure for the reproducibility of an experiment (i.e. how likely is it to obtain the same weighted mean if the same amount of zircons from the same sample are analysed again) but one of the main assumptions for using the standard error as the uncertainty of a weighted mean is that the data is normally distributed around the expected value. Due to rapid data acquisition by in-situ techniques calculated standard errors can result in

uncertainty envelopes of <0.1% for a sample. In the case of the LA-ICP-MS dates from Batu Hijau the standard error of the weighted mean (~1%: ~40 ka) is on the same order of magnitude as an individual CA-ID-TIMS date and therefore smaller than the geological spread of zircon crystallisation dates, which negates a normal distribution of the zircon data. The combination of using a weighted mean to describe a non-gaussian sample distribution with the very small attributed uncertainties results in highly precise dates that have no relation to a specific geological event.

The MSWD (A reduced chi-square statistic) of a data-set provides a first measure to indicate whether your dates are normally distributed around an expected value and thus whether the calculated weighted mean and standard error are of significance ($\sqrt{(\frac{2}{n-1})}$) rule by Wendt and Carl (1991)). As discussed before, the MSWDs for the LA-ICP-MS data are elevated, suggesting an underestimation of the individual uncertainties or, in this case, that the data do not represent a normal distribution, and thus implying that weighted means and standard errors should not be calculated to characterise a geological

event. However, the MSWDs for the SHRIMP zircon analyses of the Intermediate and Old Tonalite are acceptable, mainly due to the higher individual uncertainties. Still, they are similarly affected by protracted zircon crystallisation, which biases the weighted mean to higher values (Fig. 11). Furthermore, overestimated individual uncertainties can result in acceptable MSWDs but similarly inaccurate dates and low standard errors. For example, increasing individual uncertainties for the $^{206}$Pb-$^{238}$U dates obtained for the Old Tonalite by LA-ICP-MS by a factor of 1.5 would result in an acceptable MSWD (0.95)

but the weighted mean and standard error would be nearly identically precise but inaccurate (3.880 ± 0.041, 2SE) to those calculated with the actual uncertainties. Based on the presented data it is advised not to characterise a geological event by a weighted mean with an associated standard error if the MSWD is elevated (Wendt and Carl, 1991) and if presented it should be referred to as date and not an age. Even if the MSWD of a dataset is acceptable this is no absolute confirmation that the presented date is accurate within the presented uncertainty, unless there is evidence that the data are uniformly distributed

around the dated event or uncertainties are sufficiently high.

An attempt to obtain reliable porphyry emplacement ages from convoluted datasets could be to apply the weighted mean on the youngest or geochemically most evolved population of zircons. Differentiating zircon populations based on geochemical affinity could potentially work in situations where there are clear temporally resolved chemical trends (e.g. Wotzlaw et al., 2013;Samperton et al., 2015;Large et al., 2018). In most scenarios however, there are general geochemical





trends but they are strongly convoluted on a temporal scale (e.g Schoene et al., 2012;Rivera et al., 2014;Buret et al., 2016). For example, at Batu Hijau early crystallised zircons can have the same chemical signature (e.g. Th/U: Fig. 6) as some of the youngest zircons. In summary, identifying the youngest zircon population (e.g. by its geochemical signature) and applying a weighted mean to it could significantly increase the accuracy of the calculated emplacement age but it requires a detailed understanding of the geochemical evolution of the crystallising magma reservoir.

Based on the presented data we would recommend to use an uncertainty attributed to the weighted mean that is more representative of the uncertainty of the individual analyses, so that it will most likely cover the actually dated event. Here, we tested the standard deviation of zircons dates from each sample as a measure for the uncertainty of the weighted mean. This approach would give a more realistic estimation of the uncertainty associated with calculating a weighted mean of a data-set as it describes the variability in the measurements, (0.29 – 0.62 Ma// Fig. 7, 11) and, importantly, it would be

independent of the number of analyses. The resulting values at least for the Pliocene Batu Hijau deposit results in appropriate uncertainties for the weighted mean, as it would cover an appreciable part of the range of in-situ dates and thus the >300 kyr of zircon crystallisation and the emplacement age. Another approach would be the calculation of the dispersion of a data-set (Vermeesch, 2010, 2018) where not all data is treated as part of a single population but where the possibility of data dispersion of the analysed sample set is considered. For the presented LA-ICP-MS data-sets this would result in apparent

dispersions of 212 +43/-39 kyr, 229 +43/-39 kyr and 191 +41/-36 kyr for the data of the Old, Intermediate and Young Tonalite, similar to the actually recorded durations of zircon crystallisation. However, this approach requires a precise estimate of the associated individual uncertainties. Similar to calculating the MSWD, over- or under-estimated uncertainties would significantly modify the result.

The presented data highlights the importance of CA-ID-TIMS zircon U-Pb geochronology to resolve complicated
zircon crystallisation patterns, which in turn allow to adjust in-situ techniques. While the individual LA-ICP-MS data appear to be accurate (Fig. 11), weighted means and standard errors of high-n datasets (e.g. 3.879 ± 0.039 Ma for Old Tonalite) may provide precise mean zircon crystallisation ages but are likely to be inaccurate in determining emplacement or eruption ages if there is only a minor degree of protracted zircon crystallisation. Taking into account the dispersion of the dataset (e.g. 212 +43/-39 kyr for the Old Tonalite) or using the standard deviation (320 kyr), results in estimates of the emplacement age that

overlap with the emplacement age suggested by ID-TIMS and appear to be a more honest way of treating the data. The resulting emplacement ages may not be precise enough to resolve short timescales but can provide a timeline of broader scale magmatic events. In the case of porphyry research high-precision ID-TIMS dates are required to resolve the durations of porphyry emplacement and hydrothermal processes but in-situ data can reliably reconstruct a timeline of magma emplacement events within porphyry districts over by Myr timescales (e.g. Rezeau et al., 2016). More generally, it is

understandable that the highest possible precision is strived for from a single dataset. However, it should be refrained from increasing the precision purely by statistical measure without valid assumptions or knowledge that the boundary conditions are met as this can result in hugely precise but inaccurate dates. Furthermore, combination with in-situ petrochronology



techniques (i.e. U-Pb isotope and geochemical data from the same analyte) allows to screen zircons for inheritance and more importantly provides spatially resolved geochemical information that can be integrated with high-precision dates.

**6 Conclusions**

High-precision zircon geochronology by CA-ID-TIMS combined with in-situ zircon geochemistry provides valuable datasets that allow the reconstruction of geological processes with the highest temporal resolution. At Batu Hijau zircons record the magmatic to hydrothermal evolution of the world-class Batu Hijau porphyry Cu-Au deposit from the onset of zircon crystallisation to emplacement of the post-ore Young Tonalite. The magma reservoir that sourced the tonalites and the Cu-

Au mineralising fluids records zircon crystallisation over $336 \pm 27$ kyr. Emplacement of the first exposed tonalite at the Batu Hijau deposit (Old Tonalite) occurred after $246 \pm 28$ kyr of uninterrupted zircon crystallisation in this subjacent reservoir. Zircon trace element signatures support a dominantly crystallising and cooling magma reservoir over $285 \pm 24$ kyr until emplacement of the Intermediate Tonalite. After emplacement of the Intermediate Tonalite the chemistry of the reservoir remained in rather steady conditions for $51 \pm 28$ kyr during which it could have been disturbed by magmatic recharge or

underplating until final emplacement of the Young Tonalite. Ore formation is most probably associated with the last stages of the chemically and thermally evolving magma reservoir. The maximum duration of ore formation can be constrained to <122 kyr by the emplacement ages of pre-to syn-ore Old Tonalite and the post-ore Young Tonalite. This maximum duration of ore formation covers different pulses of mineralisation that could have lasted only a few kyr. We record a magmatic system that was active over ~250 kyr before emplacement of the first porphyry intrusion and onset of several pulses of

hydrothermal activity forming the world-class ore reserve in less than 100 kyr.

Comparison between in-situ LA-ICP-MS and SHRIMP as well as CA-ID-TIMS U-Pb geochronology reveals that all techniques provide accurate individual dates (within the stated confidence interval). However, statistical treatment of in-situ data by calculating a weighted mean and standard error can result in highly precise but inaccurate and therefore geologically meaningless ages. The tempo of magma evolution and hydrothermal processes associated with magmatic-

hydrothermal systems, such as porphyry deposits is too fast to be reliably resolved by in-situ U-Pb geochronology and requires ID-TIMS geochronology. Combination of high-precision geochronology with in-situ or TIMS-TEA geochemistry is currently the most powerful tool in deciphering these geologically rapid processes.

**7 Data availability**

All data used in this manuscript is available from the supplementary files.



# 8 Supplement link

# 9 Author contribution

Simon Large conducted the ID-TIMS measurements. Simon Large together with Marcel Guillong conducted the LA-ICP-MS measurements. The study was designed by Simon Large, Christoph Heinrich, Albrecht von Quadt and Jörn-Frederik
Wotzlaw. Discussion of the data involved all authors and Simon Large wrote the manuscript and drafted the figures with input from all co-authors

# 10 Acknowledgement

This work was supported through the Swiss National Science Foundation project 200026-166151. Wotzlaw acknowledges funding through the ETH Zurich Postdoctoral Fellowship Program. Extensive logistical support by the Geology Department
of the Batu Hijau mine, especially Eddy Priowasono, and the technical staff at the mine site was hugely appreciated.

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
