# Peer review of "Resolving the timescales of magmatic and hydrothermal processes associated with porphyry deposit formation using zircon U-Pb petrochronology"

_Geochronology, 2020_

## Referee Comment (RC1) · Fernando Corfu (Referee) · 28 Apr 2020

**Review of paper:  gchron-2020-5:**

**Resolving the timescales of magmatic and hydrothermal processes associated with porphyry deposit formation using zircon U-Pb petrochronology**
Simon J.E. Large, Jörn F. Wotzlaw, Marcel Guillong, Albrecht von Quadt, Christoph A. Heinrich

The paper reports an excellent set of data for zircon from several tonalite phases encompassing the time of formation of a major Cu-Au deposit in Indonesia. The results constrain the period of formation of zircon in the different magma batches, and the time of formation of the mineralization, also providing information of the magma evolution based on trace element data. The analytical results are of very good quality and the interpretation is overall reasonable. The paper is well prepared with just few typos or other problems. I have put a number of questions and suggestions on specific details directly in the file.

Here some comments on aspects of the paper that the authors should consider.

The first concerns the definition of the 'age of emplacement'. The nice thing with modern ID-TIMS U-Pb data is the high time resolution it achieves, which permits to separate out and date very specific segments of geological processes. This advantage, however, brings new challenges requiring to be more specific on the definitions of the specific parts of the process that are dated. For example, on line 304-305 it states: '... zircon dates of $3.736 \pm 0.023$ Ma, $3.697 \pm 0.018$ Ma and $3.646 \pm 0.022$ Ma …[are interpreted] as the time of respective porphyry emplacement …' which concludes a period of >200 ka of zircon crystallization. This implies that the last zircon in each rock crystallized just as the magma reached its present position. One could wonder why the last zircon couldn't have crystallized well before the magma reached this final position, or alternatively much later than the emplacement. Some information in merit is provided much later in the Discussion, but clearly, these interpretations and the arguments are quite fundamental in such a paper, and need to be presented before anything else.
A somewhat related problem concerns the question of the validity of the results. I am impressed by the high quality of the data, the superb blanks and the high precision. Nevertheless, a central factor in all data sets is the reproducibility of individual analyses. You are measuring 0.5 pg of Pb, next to nothing, and still achieve a very good precision. But is the precision identical with, or less good than the reproducibility of such measurements? To substantiate the solidity of the work, the authors should present information that backs up the implication at the zircon age of each individual zircon grain is reproducible, the alternative being that the larger spread of the ages in each sample may represent a closer measure of the reproducibility. I have seen some very good such data sets in other papers that support their validity, but the question is central with every new application and needs to be addressed.

My third main point concerns the subsidiary part of the paper, which discussed the comparison of ID-TIMS with ICP and SIMS data and reflects on their applications, correctness, and statistical factors. I find this parts absolutely atrocious, and I highly recommend to cut it out. The ICP analyses of these young zircons achieve intensities of maybe 100 cps for mass 206, for measurements lasting less than a minute, and there is no indication that it even gets to evaluate things like the need to correct for common Pb. So, the results are of very low precision, and it is a wonder that they are even close to the real values. The SIMS data are more substantial, but also they face incredible measurement challenges. So, really, all the arguments on statistics and processes cannot get around these basic limitations. And talking about them to such an extent  is like watching children playing in the sand. Boring. Suggest cutting this parts out, keeping them for some contribution to a technical workshop, and not use them to spoil an otherwise interesting paper.

April 2nd, 2020   F. Corfu

[revised manuscript text omitted]

---

## Referee Comment (RC2) · Brenhin Keller (Referee) · 10 May 2020

I apologize for the lateness of this review, and hope that it will still be of use.

The CA-ID-TIMS dataset presented by Large et al. is impressive, and contributes significantly to our understanding of the timescale and tempo of economically significant porphyry associated magmatism. The reported analytical precision is excellent for grains of this young age, and the analytical techniques suggest confidence that this precision is backed up by equivalent accuracy. Among other points, the CL imaging

and in-situ geochemical characterization of each dated grain is to be applauded.

My main point of discussion involves the use of single "oldest" and "youngest" zircons to constrain the duration of zircon crystallization and metal precipitation. For the particular regime the authors are working in (N $\sim$ 15, apparent $\Delta t \sim$ 10-20$\sigma$), the competing effects of undersampling and analytical dispersion likely mostly cancel. On such a basis, the authors could perhaps argue to continue with this approach if they wish. However, "oldest/youngest zircon" is still not inherently statistically robust. One general solution (to which I am obviously biased) would be that of doi:10.7185/geochemlet.1826 (if you go this route, I would probably suggest a uniform $\vec{f}_{xtal}$) – but my own work is certainly not the only option here. As I understand it, Pieter Vermeesch also has a perfectly workable analytical minimum age calculator (effectively based on an assumption of a truncated normal $\vec{f}_{xtal}$) in IsoplotR. In either case, it will not materially affect the major conclusions of the study.

While I can see the previous reviewer's point that the in-situ data could be cut since they are so imprecise, it also seems that this data is critical proof of the authors' claim that, at the very least for the Batu Hijau porphyry-Cu-Au deposit, "geologically rapid events or processes or the tempo of magma evolution are too fast to be reliably resolved by in-situ U-Pb geochronology and require ID-TIMS geochronology." Consequently, I would leave it up to the authors which way they wish to proceed on this front.

———-

52-63: This may be somewhat overoptimistic; there is a substantial literature on hydrothermal alteration of zircon in both lab and field contexts. "Resistant" might be more accurate.

553: "petrochronological"

---

## Author Comment (AC2) · 1 Jun 2020

We thank the B. Keller for his constructive and positive comments on our submitted manuscript. We provide answers to the reviewer's main comments below.

1. Stochastical sampling approach to determine emplacement ages

We thank the reviewer for pointing us towards stochastical sampling approach to determine porphyry emplacement ages. We have calculated emplacement ages based on this approach using the interactive Jupiter notebook on

https://github.com/brenhinkeller/BayeZirChron.c. We have addressed the results in the discussion (Section 5.5) and have added the emplacement ages to the appendix. Indeed, the different treatments of the CA-ID-TIMS result in overlapping results with little variation. More importantly the durations and timescales remain nearly identical.

2. Discussion TIMS vs. in-situ data

We thank the reviewer for this assessment. Indeed, highlighting the differences in apparent and absolute resolution between in-situ and ID-TIMS geochronology is the main point of the later discussion. We hope to provide a contribution to the scientific literature by providing a data-set where the differences can be investigated from analyses of zircons from the same samples. Thus, we would like to leave the shortened and focussed discussion in the manuscript.

Minor comments were addressed accordingly

Best regards Simon Large et al.

---

## Editor Comment (EC1) · Daniela Rubatto (Editor) · 3 Jun 2020

Please add a short paragraph in the main text to justify the reproducibility of young CA-ID-TIMS data, along the lines of what stated in the reply.
* * *

---

## Author Response (AR1)

**Replies to Reviewer #1 of paper: gchron-2020-5:**

**Resolving the timescales of magmatic and hydrothermal processes associated with porphyry deposit formation using zircon U-Pb petrochronology**
Simon J.E. Large, Jörn F. Wotzlaw, Marcel Guillong, Albrecht von Quadt, Christoph A. Heinrich

We thank F. Corfu for his detailed comments on our submitted manuscript. We provide replies (blue) to the main comments of the reviewer and have addressed all of his valid points in the annotated manuscript.

(1) The first concerns the definition of the 'age of emplacement'. The nice thing with modern ID-TIMS UPb data is the high time resolution it achieves, which permits to separate out and date very specific segments of geological processes. This advantage, however, brings new challenges requiring to be more specific on the definitions of the specific parts of the process that are dated. For example, on line 304-305 it states: '… zircon dates of 3.736 ± 0.023 Ma, 3.697 ± 0.018 Ma and 3.646 ± 0.022 Ma …[are interpreted] as the time of respective porphyry emplacement …' which concludes a period of >200 ka of zircon crystallization. This implies that the last zircon in each rock crystallized just as the magma reached its present position. One could wonder why the last zircon couldn't have crystallized well before the magma reached this final position, or alternatively much later than the emplacement. Some information in merit is provided much later in the Discussion, but clearly, these interpretations and the arguments are quite fundamental in such a paper, and need to be presented before anything else.

> We thank the reviewer for the valid point that the reasoning for the interpretation of the youngest U-Pb dates as the porphyry emplacement ages should be made earlier. As pointed out in the manuscript annotated by the reviewer it was previously in lines 578 – 579 in the methods section. We have added the reasoning with a bit more detail to section 4.3 "CA-ID-TIMS geochronology".
> Porphyry intrusions are volumetrically minor sub-volcanic intrusions and are considered to cool rapidly upon emplacement. It is thus assumed that most, if not all, zircons crystallise in the underlying magma reservoir resulting in the extended timescales of zircon crystallisation. Zircon is considered a low temperature phase crystallising until reaching the solidus the youngest recorded zircon and thus records the full crystallisation of the intrusions. The individual uncertainty of a zircon date is sufficient to account for the timescales of porphyry cooling (<10 kyr; e.g. Cathles, 1977).

(2) A somewhat related problem concerns the question of the validity of the results. I am impressed by the high quality of the data, the superb blanks and the high precision. Nevertheless, a central factor in all data sets is the reproducibility of individual analyses. You are measuring 0.5 pg of Pb, next to nothing, and still achieve a very good precision. But is the precision identical with, or less good than the reproducibility of such measurements? To substantiate the solidity of the work, the authors should present information that backs up the implication at the zircon age of each individual zircon grain is reproducible, the alternative being that the larger spread of the ages in each sample may represent a closer measure of the reproducibility. I have seen some very good such data sets in other papers that support their validity, but the question is central with every new application and needs to be addressed.

> We thank the reviewer for outlining the high quality of the presented data. Indeed, we are measuring very low quantities of Pb. However, we are convinced that the results are reproducible. The zircon standards referred to and referenced in the manuscript (Aus_Z7_5, von Quadt et al., 2016) contain similarly low amounts of radiogenic Pb (0.5 – 4 pg), are of similar age (2.41 Ma) and most importantly are reproducible on the kyr scale.. These were analysed over the same time interval, under the same conditions and in the same lab as the zircons analysed for this study. And these measurements

A more indirect assurance that our data is reproducible is that timescales of zircon crystallisation appear to be remarkably similar for studies on porphyry deposits not only conducted in this lab, but also in other labs (Fig. 8). These studies have been conducted on deposits of variable age and thus on zircons with hugely variable radiogenic Pb contents. Finally, the sequence of youngest ages matches the geological emplacement sequence, despite the much larger range of older zircons in each sample; an observation that holds for every single case of several other deposits we studied with this approach. This systematics would be impossible to explain if the age variations were analytical artefacts, and it a also corroborates our interpretation of the youngest zircon in each sample being close to the age of emplacement and magma quenching and termination of zircon crystallization.

(2) My third main point concerns the subsidiary part of the paper, which discussed the comparison of IDTIMS with ICP and SIMS data and reflects on their applications, correctness, and statistical factors. I find this parts absolutely atrocious, and I highly recommend to cut it out. The ICP analyses of these young zircons achieve intensities of maybe 100 cps for mass 206, for measurements lasting less than a minute, and there is no indication that it even gets to evaluate things like the need to correct for common Pb. So, the results are of very low precision, and it is a wonder that they are even close to the real values. The SIMS data are more substantial, but also they face incredible measurement challenges. So, really, all the arguments on statistics and processes cannot get around these basic limitations. And talking about them to such an extent is like watching children playing in the sand. Boring. Suggest cutting this parts out, keeping them for some contribution to a technical workshop, and not use them to spoil an otherwise interesting paper.

We appreciate the reviewer outlining the low precision of the individual LA-ICP-MS dates and SHRIMP dates. We also appreciate the positivity of Reviewer #2 regarding this section. The main point of this section is not that the LA-ICP-MS dates further define the geological interpretation, in which case they could be considered unnecessary. However, we compare the three most commonly applied U-Pb techniques on zircons from the same samples and in the case of TIMS and LA-ICP-MS on the same zircon grains. From this comparison (of a type that has, to our knowledge, not been published before), we can show (1) that the large number of young low precision LA-ICP-MS dates is remarkably accurate as a bulk data-set but that (2) the calculation of the weighted mean and the standard error from LA-ICP-MS populations as currently practiced in the copious literature needs more careful consideration. Using weighted means on such young data-sets as estimates for any geological event or process can result in highly precise dates without any geological significance. We therefore believe that this discussion is of interest to a broad readership. We tried to shorten the section and to focus on our main points and hope that the reviewer and editor can accept the revised section.

With best regards,
Simon Large et al.

**Resolving the timescales of magmatic and hydrothermal processes associated with porphyry deposit formation using zircon U-Pb petrochronology**
Simon J.E. Large, Jörn F. Wotzlaw, Marcel Guillong, Albrecht von Quadt, Christoph A. Heinrich

We thank the Brenhin Keller for his constructive and positive comments on our submitted manuscript. We provide answers to the reviewer's main comments below. The reviewer's original comments are in black and our responses in blue.

**Brenhin Keller (Referee)**
cbkeller@dartmouth.edu

I apologize for the lateness of this review, and hope that it will still be of use.
The CA-ID-TIMS dataset presented by Large et al. is impressive, and contributes significantly to our understanding of the timescale and tempo of economically significant porphyry associated magmatism. The reported analytical precision is excellent for grains of this young age, and the analytical techniques suggest confidence that this precision is backed up by equivalent accuracy. Among other points, the CL imaging and in-situ geochemical characterization of each dated grain is to be applauded.

1.    My main point of discussion involves the use of single "oldest" and "youngest" zircons to constrain the duration of zircon crystallization and metal precipitation. For the particular regime the authors are working in (N _ 15, apparent _t _ 10-20_), the competing effects of undersampling and analytical dispersion likely mostly cancel. On such a basis, the authors could perhaps argue to continue with this approach if they wish. However, "oldest/youngest zircon" is still not inherently statistically robust. One general solution (to which I am obviously biased) would be that of doi:10.7185/geochemlet.1826 (if you go this route, I would probably suggest a uniform ~ $f_{xtal}$) – but my own work is certainly not the only option here. As I understand it, Pieter Vermeesch also has a perfectly workable analytical minimum age calculator (effectively based on an assumption of a truncated normal ~ $f_{xtal}$) in IsoplotR. In either case, it will not materially affect the major conclusions of the study.

We thank the reviewer for pointing us towards stochastical sampling approach to determine porphyry emplacement ages. We have calculated emplacement ages based on this approach using the interactive Jupiter notebook on https://github.com/brenhinkeller/BayeZirChron.c. We have addressed the results in the discussion (Section 5.5: lines 619- 622) and have added the emplacement ages to the appendix. Indeed, the different treatments of the CA-ID-TIMS result in overlapping results with little variation. More importantly the durations and timescales remain nearly identical.

2.    While I can see the previous reviewer's point that the in-situ data could be cut since they are so imprecise, it also seems that this data is critical proof of the authors' claim that, at the very least for the Batu Hijau porphyry-Cu-Au deposit, "geologically rapid events or processes or the tempo of magma evolution are too fast to be reliably resolved by insitu U-Pb geochronology and require ID-TIMS geochronology." Consequently, I would leave it up to the authors which way they wish to proceed on this front.

We thank the reviewer for this assessment. Indeed, highlighting the differences in apparent and absolute resolution between in-situ and ID-TIMS geochronology is the main point of the later discussion. We hope to provide a contribution to the scientific literature by providing a data-set where the differences can be investigated from analyses of zircons from the same samples. Thus, we would like to leave the shortened and focussed discussion in the manuscript.

Minor comments were addressed accordingly

Best regards

Simon Large et al.

[revised manuscript text omitted]

---

## Author Response (AR2)

Dear Daniela Rubatto,

Thank you for making the points on our Supplementary tables.

- IDTIMS dates: negative values for 207/206 ages are not acceptable. If no reliable date can be calculated, then no value should be reported.

Although, some of the 207/206 dates were negative they were attributed with large uncertainties due to the low amounts of 207Pb measured in these young zircons. We agree that reporting those dates is confusing and we have deleted them from the tables (TIMS and LA-ICP-MS). We have also excluded 207/235 LA-ICP-MS dates due to the same concern. If a reader is interested in these ages they can easily calculate them with the provided data.

- LAICPMS dates: negative values for error correlation are likely a mistake, Please check the formula used for generating this value. Same issue as above for negative ages.

The negative ratios are not a mistake but the result of relatively large uncertainties associated with the 207/235 and 206/207 ratios (see formulas 118 and 122 in Schmitz and Schoene, 2007). Values for error correlations can theoretically range from 1 to -1. We would thus like to keep the negative error correlations in the manuscript.

- LAICPMS dates: please ensure that all values in the same column have the same N of digits and (0.000027 versus 1). "0" is not an acceptable value for an isotopic ratio, probably one isotope is below detection.

- TE Table: watch the significant digits, more than 3 digits (e.g. 1.597) imply precision better than 1%, which is certainly not the case. Some values have only one digit, probably they need more.

We have tried to address these valid points. We tried to keep the same number of values per column. We did not modify the numbers as such but just adjusted the number of illustrated digits. In this way the reader is able to examine the number of digits he wishes and re-calculate data with the original numbers.

We hope that these adjustments are satisfactory and please let us know if anything else is required changing.

Thank you and best regards

Simon Large et al.